# E-Bayesian and H-Bayesian Inferences for a Simple Step-Stress Model with Competing Failure Model under Progressively Type-II Censoring

**DOI:** 10.3390/e24101405

**Published:** 2022-10-01

**Authors:** Ying Wang, Zaizai Yan, Yan Chen

**Affiliations:** 1College of Science, Inner Mongolia University of Technology, Hohhot 010051, China; 2School of Statistics and Mathematics, Inner Mongolia University of Finance and Economics, Hohhot 010070, China; 3Institute of Mathematics and Statistics, Wuhan University, Wuhan 430072, China

**Keywords:** step-stress accelerated life test, competing risk model, cumulative exposure model, Bayesian estimate, expected Bayesian estimation, hierarchical Bayesian estimation

## Abstract

In this paper, we discuss the statistical analysis of a simple step-stress accelerated competing failure model under progressively Type-II censoring. It is assumed that there is more than one cause of failure, and the lifetime of the experimental units at each stress level follows exponential distribution. The distribution functions under different stress levels are connected through the cumulative exposure model. The maximum likelihood, Bayesian, Expected Bayesian, and Hierarchical Bayesian estimations of the model parameters are derived based on the different loss function. Based on Monte Carlo Simulations. We also get the average length and the coverage probability of the 95% confidence intervals and highest posterior density credible intervals of the parameters. From the numerical studies, it can be seen that the proposed Expected Bayesian estimations and Hierarchical Bayesian estimations have better performance in terms of the average estimates and mean squared errors, respectively. Finally, the methods of statistical inference discussed here are illustrated with a numerical example.

## 1. Introduction

In recent years, it has been challenging to obtain sufficient failure data during the general service condition with increasing reliability of products. Accelerated life testing (ALT) is conducted to overcome such difficulties. In ALT, the test units are subjected to higher levels of stress condition for shortening the total testing time, and sufficient failure data can be obtained for reliability assessment. Some authentic books in the field of the ALT include Nelson et al. [1], Nelson [2], Meeker et al. [3], and Bagdonavicius et al. [4]. The constant-stress accelerated life testing (C-SALT) and the step-stress accelerated life testing (S-SALT) are two special types of ALT. S-SALT an advantage of yielding more failure data in a limited testing time and changes the stress level at a prefixed time or a prefixed number of failures during the testing. To analyze the data from S-SALT, one requires a model that relates to the failure lifetimes under different stress levels. The cumulative exposure model (CEM) is the most studied in the literature, and the model was first introduced by Sedyakin [5]. The S-SALT under the assumption of CEM has attracted great attention, such as Balakrishnan et al. [6,7], Lee et al. [8], and Tang [9]. Furthermore, it is common that the products’ failure may be caused by more than one cause. Therefore, each of these causes would compete with each other to result in the final failure. We call those possible failure causes competing failures. The competing failure model plays an important role and the competing failure data are analyzed by Cox [10] and David et al. [11]. In S-SALT, some researchers have discussed the competing failure model, such as Balakrishnan et al. [12], Beltrami [13,14], Shi and Liu [15], Srivastava et al. [16], Xu et al. [17], Zhang et al. [18,19], Ganguly et al. [20], Han et al. [21,22,23], Varhgese et al. [24], Liu et al. [25], Abu-Zinadah et al. [26] and Aljohaniet al. [27]. The maximum likelihood estimation (MLE) and the Bayesian estimation (BE) based on the different loss function (LF) are the common inference for analyzing statistical data. Lindley [28] introduced the Hierarchical Bayesian estimation (H-BE) primarily, and it was examined by Han [29]. The Expected Bayesian estimation (E-BE) is the expectation of BE, and it was introduced by Han [30]. Han [31,32] derived the E-BE and H-BE of the reliability parameter under testing data. However, there are few works concerning the E-Bayesian and H-Bayesian inference for the step-stress accelerated competing failure model.

We will discuss the EB and HB inference for a simple step-stress accelerated competing failure model which has only two stress levels, and the layout of the paper is organized as follows. Section 2 gives the model and basic assumptions. Asymptotic confidence intervals (CIs) and Bootstrap confidence intervals (BCIs) are constructed in Section 3. Bayesian estimations (BEs) are derived based on the different LFs in Section 4. The E-BE is derived based on the different LFs in Section 5. The H-BEs is derived based on the different LFs in Section 6. The Simulation data analysis is provided in Section 7. Section 8 illustrates the proposed methods by a numerical example. Finally, we provide concluding remarks and future research proposes.

## 2. Model Description and MLEs

In this section, we discuss a simple-stress life testing and provide the MLEs of the unknown parameters based the observed data.

### 2.1. Basic Assumption

To describe the simple S-SALT clearly, some assumptions are made as follows:

(1)The unit fails only due to one of the two independent competing failure causes with failure times $T_{1}$ and $T_{2}$, respectively. A failure time is recorded as joint random variable $\left(T,\delta \right)$. Let $\delta =\left\{\begin{array}{l} 1,\mathrm{the}\;\mathrm{failure}\;\mathrm{is}\;\mathrm{caused}\;\mathrm{by}\;\mathrm{the}\;\mathrm{first}\;\mathrm{cause}\\ 2,\mathrm{the}\;\mathrm{failure}\;\mathrm{is}\;\mathrm{caused}\;\mathrm{by}\;\mathrm{the}\;\sec\mathrm{ond}\;\mathrm{cause} \end{array}\right.$, that denote the indicator variable for the cause of failure time $T=\min(T_{1},T_{2})$.(2)The lifetime follows an exponential distribution with scale parameter $\lambda_{ij}$. Let $1/\lambda_{ij}$ be the mean time-to-failure of a test unit at the stress level $S_{i}$ by the failure cause j for i,j=1,2. The cumulative distribution function (CDF) and probability density function (PDF) are given as follows, respectively
(1)$$F_{ij}(t)=F(t;\lambda_{ij})=1-\exp(-\lambda_{ij}t)$$
(2)$$f_{ij}(t)=f(t;\lambda_{ij})=\lambda_{ij}\exp(-\lambda_{ij}t)$$
where t≥0, $\lambda_{ij}>0$ and i,j=1,2.(3)The scale parameter $\lambda_{ij}$ agrees with a log-linear function of stress
(3)$$\ln\lambda_{ij}=a_{j}+b_{j}\varphi (S_{i})$$
where $a_{j}$ and $b_{j}$ are unknown coefficient parameters, $\varphi (S_{i})=1/S_{i}$ that is chosen as the Arrhenius model [2] in this paper, i,j=1,2, but this not used explicitly in this paper.(4)Lifetime distribution at different stress levels is related by CEM. The CEM assumes that the remaining lifetime of a unit depends only on the cumulative exposure accumulated at the current stress level, regardless of how the exposure is actually accumulated. The failure probability of product working time $t_{1}$ under stress $S_{1}$ is equivalent to the failure probability of product working time $t_{2}$ under stress $S_{2}$. At the time τ when the stress level increases from $S_{1}$ to $S_{2}$, the CDF of the lifetime of the test unit failed due to cause j for j=1,2 can be written as follows:(4)$$F_{j}\left(t\right)=\left\{\begin{array}{cc} F_{1j}(t), & \;\;0<t<\tau \\ F_{2j}(t-a_{j}), & \;\;t>\tau \end{array}\right.$$
where $F_{1j}$ and $F_{2j}$ are the CDFs of the lifetime of the test unit failed under stress $S_{1}$ and $S_{2}$, respectively, and $a_{j}$ is such that it satisfies $F_{1j}(\tau )=F_{2j}(\tau -a_{j})$ and $a_{j}=\left(1-\lambda_{1j}/\lambda_{2j}\right)\tau$.

Under these assumptions, the CDF of the lifetime of the test unit failed due to cause j is
(5)$$F_{j}(t)=F_{j}(t;\lambda_{1j},\lambda_{2j})=\left\{\begin{array}{cc} 1-\exp(-\lambda_{1j}t), & \;\;\;0<t<\tau \\ 1-\exp\left(-(\lambda_{1j}-\lambda_{2j})\tau -\lambda_{2j}t\right), & \;t>\tau \end{array}\right.$$
and the corresponding PDF is given by
(6)$$f_{j}(t)=f_{j}\left(t;\lambda_{1j},\lambda_{2j}\right)=\left\{\begin{array}{cc} \lambda_{1j}\exp(-\lambda_{1j}t), & \;\;\;0<t<\tau \\ \lambda_{2j}\exp\left(-(\lambda_{1j}-\lambda_{2j})\tau -\lambda_{2j}t\right), & \;t>\tau \end{array}\right.$$
where j=1,2.

### 2.2. Model Description

Under a progressively Type-Ⅱ censored (PT-II-C) scheme, the simple S-SALT is described as follows:

The n test units are placed in the test under the initial stress level $S_{1}$. At the first failure time $t_{1:n}$, $R_{1}$ units are progressively removed from the remaining n−1 units and recording data $\left(t_{1:n},\delta_{1},R_{1}\right)$. The test continues until time $t_{N_{1}:n}$, $R_{N_{1}}$ units are progressively removed and recording data $\left(t_{N_{1}:n},\delta_{N_{1}},R_{N_{1}}\right)$. Then, we increased the stress level to $S_{2}$, and the remaining $\left(n-N_{1}-R_{1}-\cdots -R_{N_{1}}\right)$ units continued to be tested. At the time of $(N_{1}+1)\mathrm{th}$ failure, $R_{N_{1}+1}$ units were progressively removed, and we got the sample $\left(t_{N_{1}+1:n},\delta_{N_{1}+1},R_{N_{1}+1}\right)$. The test is continued until the $(N_{1}+N_{2})\mathrm{th}$ failure is observed, $R_{N_{1}+N_{2}}$ units are removed, and the test terminates. Here, $N_{1}$, $N_{2}$, $R_{1},\cdots ,R_{N_{1}+N_{2}}$$\left(N_{1}+N_{2}+R_{1}+\cdots +R_{N_{1}+N_{2}}=n\right)$ are prefixed. Therefore, under PT-II-C scheme, the observed data for the simple S-SALT are that:$$S_{1}:(t_{1:n},\delta_{1},R_{1}),(t_{2:n},\delta_{2},R_{2}),\cdots ,(t_{N_{1}:n},\delta_{N_{1}},R_{N_{1}});\;S_{2}:(t_{N_{1}+1:n},\delta_{N_{1}+1},R_{N_{1}+1}),(t_{N_{1}+2:n},\delta_{N_{1}+2},R_{N_{1}+2}),\cdots ,(t_{N_{1}+N_{2}:n},\delta_{N_{1}+N_{2}},R_{N_{1}+N_{2}}).$$

Here, $t_{1:n},\cdots ,t_{N_{1}+N_{2}:n}$ are order statistics, $\delta_{i}\in \{1,2\},i=1,2,\cdots ,N_{1}+N_{2}$.

### 2.3. Maximum Likelihood Estimates

Based on the assumptions (1)–(4) and the lifetime $T=\min(T_{1},T_{2})$ of the test unit, the CDF and PDF of T can be obtained as follows:(7)$$\begin{array}{cc} F_{T}(t) & =1-\prod_{j=1}^{2}\left(1-F_{j}(t)\right)\\ & =\left\{\begin{array}{cc} 1-\exp\left(-(\lambda_{11}+\lambda_{12})t\right), & 0\leq t\leq \tau \\ 1-\exp\left(-(\lambda_{11}+\lambda_{12}-\lambda_{21}-\lambda_{22})\tau -(\lambda_{21}+\lambda_{22})t\right), & t>\tau \end{array}\right. \end{array}$$
(8)$$f_{T}(t)=\left\{\begin{array}{cc} (\lambda_{11}+\lambda_{12})\exp\left(-(\lambda_{11}+\lambda_{12})t\right), & 0\leq t\leq \tau \\ (\lambda_{21}+\lambda_{22})\exp\left(-(\lambda_{11}+\lambda_{12}-\lambda_{21}-\lambda_{22})\tau -(\lambda_{21}+\lambda_{22})t\right),\;t>\tau \end{array}\right.$$

Then the joint distribution of (T,δ) is given by
(9)$$\begin{array}{cc} f_{T,\delta }(t,j) & =f_{j}(t)\left(1-F_{j^{*}}(t)\right)\\ & =\left\{\begin{array}{cc} \lambda_{1j}\exp\left(-(\lambda_{11}+\lambda_{12})t\right), & 0\leq t\leq \tau \\ \lambda_{2j}\exp\left(-(\lambda_{11}+\lambda_{12}-\lambda_{21}-\lambda_{22})\tau -(\lambda_{21}+\lambda_{22})t\right),\;t>\tau \end{array}\right. \end{array}$$
where $j,j^{*}=1,2$ and $j\neq j^{*}$.

With the life-testing scheme described above, the following ordered failure time will be observed:$$\left\{t_{1:n}<t_{2:n}<\cdots <t_{N_{1}:n}<t_{N_{1}+1:n}<\cdots <t_{N_{1}+N_{2}:n}\right\}$$

$n_{1j}$ is the number of units that fail under stress $S_{1}$ due to the failure cause j; $n_{2j}$ is the number of units that fail under stress $S_{2}$ due to the failure cause j; for j=1,2 such that $N_{1}=n_{11}+n_{12}$ and $N_{2}=n_{21}+n_{22}$. The observed failure times $t=(t_{1:n},t_{2:n},\cdots ,t_{N_{1}:n},t_{N_{1}+1:n},\cdots ,t_{N_{1}+N_{2}:n})$ and the $\Theta =\left(\lambda_{11},\lambda_{12},\lambda_{21},\lambda_{22}\right)$. Then, the likelihood function can be written as
(10)$$\begin{array}{l} L(\Theta |t)\propto \prod_{i=1}^{N_{1}}\left[f_{T,\delta }(t_{i:n}){\left(1-F_{T}(t_{i:n})\right)}^{R_{i}}\right]\prod_{i=N_{1}+1}^{N_{1}+N_{2}}\left[f_{T,\delta }(t_{i:n}){\left(1-F_{T}(t_{i:n})\right)}^{R_{i}}\right]\\ \propto \left\{\prod_{i=1}^{2}\prod_{j=1}^{2}{\left(\lambda_{ij}\right)}^{n_{ij}}\right\}\exp\{-(\lambda_{11}+\lambda_{12})(T_{1}+T_{21})-(\lambda_{21}+\lambda_{22})T_{22}\} \end{array}$$
where $T_{1}=\sum_{i=1}^{N_{1}}\left(1+R_{i}\right)t_{i:n}$,$T_{21}=\sum_{i=N_{1}+1}^{N_{1}+N_{2}}\left(1+R_{i}\right)\tau$,$T_{22}=\sum_{i=N_{1}+1}^{N_{1}+N_{2}}\left(1+R_{i}\right)\left(t_{i:n}-\tau \right)$, and $\tau =t_{N_{1}:n}$.

By using Equation (10), the log-likelihood function can be written as:(11)$$l=\sum_{i=1}^{2}\sum_{j=1}^{2}n_{ij}\ln\lambda_{ij}-\left(\lambda_{11}+\lambda_{12}\right)\left(T_{1}+T_{21}\right)-\left(\lambda_{21}+\lambda_{22}\right)T_{22}$$ Taking the first partial derivative of Equation (11) with respect to $\lambda_{ij}$, and let it equal zero
(12)$$\left\{\begin{array}{l} \frac{\partial l}{\partial \lambda_{11}}=\frac{n_{11}}{\lambda_{11}}-\left(T_{1}+T_{21}\right)=0\\ \frac{\partial l}{\partial \lambda_{12}}=\frac{n_{12}}{\lambda_{12}}-\left(T_{1}+T_{21}\right)=0\\ \frac{\partial l}{\partial \lambda_{21}}=\frac{n_{21}}{\lambda_{21}}-T_{22}=0\\ \frac{\partial l}{\partial \lambda_{22}}=\frac{n_{22}}{\lambda_{22}}-T_{22}=0 \end{array}\right.$$ Using simple algebra calculations, we obtain
(13a)$${\hat{\lambda }}_{1j(MLE)}=\frac{n_{1j}}{T_{1}+T_{21}}$$
(13b)$${\hat{\lambda }}_{2j(MLE)}=\frac{n_{2j}}{T_{22}}$$

## 3. Interval Estimations

In this section, we propose the asymptotic confidence intervals (ACIs) and the bootstrap confidence intervals (BCIs) for the parameters $\lambda_{ij}$.

### 3.1. Asymptotic Confidence Intervals (ACIs)

According to the asymptotic likelihood theory, we get the information matrix of Θ, the elements of which are
(14)$$\begin{array}{l} I_{11}={\left.-\frac{\partial^{2}l}{\partial \lambda_{11}^{2}}\right|}_{\lambda_{11}={\hat{\lambda }}_{11}}=\frac{n_{11}}{{\hat{\lambda }}_{11}^{2}}\\ I_{22}={\left.-\frac{\partial^{2}l}{\partial \lambda_{12}^{2}}\right|}_{\lambda_{12}={\hat{\lambda }}_{12}}=\frac{n_{12}}{{\hat{\lambda }}_{12}^{2}}\\ I_{33}={\left.-\frac{\partial^{2}l}{\partial \lambda_{21}^{2}}\right|}_{\lambda_{21}={\hat{\lambda }}_{21}}=\frac{n_{21}}{{\hat{\lambda }}_{21}^{2}}\\ I_{44}={\left.-\frac{\partial^{2}l}{\partial \lambda_{22}^{2}}\right|}_{\lambda_{22}={\hat{\lambda }}_{22}}=\frac{n_{22}}{{\hat{\lambda }}_{22}^{2}}\\ I_{ij}=I_{ji}=0\left(i\neq j=1,2,3,4\right) \end{array}$$

Then, the observed Fisher information matrix of Θ is that
(15)$$\hat{I}(\Theta )=\left[\begin{array}{ccc} I_{11} & \cdots & I_{14}\\ \vdots & \ddots & \vdots \\ I_{41} & \cdots & I_{44} \end{array}\right]$$

The approximate asymptotic variance-covariance matrix can be given by $\hat{I}{\left(\Theta \right)}^{-1}$ and denoting $\hat{V}(\Theta )=\hat{I}{\left(\Theta \right)}^{-1}=\mathrm{Diag}(\frac{{\hat{\lambda }}_{11}^{2}}{n_{11}},\frac{{\hat{\lambda }}_{12}^{2}}{n_{12}},\frac{{\hat{\lambda }}_{21}^{2}}{n_{21}},\frac{{\hat{\lambda }}_{22}^{2}}{n_{22}})$. Therefore, the $100\left(1-\gamma \right)\%$ ACIs for the parameter $\lambda_{ij}$ are
(16)$$\left({\hat{\lambda }}_{ij}-Z_{\gamma /2}\frac{{\hat{\lambda }}_{ij}}{\sqrt{n_{ij}}},\;{\hat{\lambda }}_{ij}+Z_{\gamma /2}\frac{{\hat{\lambda }}_{ij}}{\sqrt{n_{ij}}}\right)$$
where $Z_{\gamma /2}$ is the γ/2th upper percentile of the standard normal distribution.

### 3.2. Bootstrap Confidence Intervals (BCIs)

#### 3.2.1. Bootstrap-p Method

Efron (1982) proposed a Bootstrap-p (Percentile bootstrap) method, and the bootstrap-p samples are generated as follows:

**Step 1****.** Given $n,N_{1},N_{2}$ and progressive censored scheme $\left(R_{1},\cdots ,R_{N_{1}+N_{2}}\right)$, compute MLEs ${\hat{\lambda }}_{ij}$ of unknown parameters $\lambda_{ij}(i,j=1,2)$ based on the PT-IIC data $(t_{1:n},\delta_{1},R_{1}),\cdots ,(t_{N_{1}:n},\delta_{N_{1}},R_{N_{1}}),$$(t_{N_{1}+1:n},\delta_{N_{1}+1},R_{N_{1}+1}),\cdots ,$$\left(t_{N_{1}+N_{2}:n},\delta_{N_{1}+N_{2}},R_{N_{1}+N_{2}}\right)$.

**Ste****p 2****.** Generated a bootstrap sample $(t_{1:n}^{*},\delta_{1}^{*},R_{1}^{*}),\cdots ,(t_{N_{1}:n}^{*},\delta_{N_{1}}^{*},R_{N_{1}}^{*}),$$(t_{N_{1}+1:n}^{*},\delta_{N_{1}+1}^{*},$$,R_{N_{1}+1}^{*})$$,\cdots ,\left(t_{N_{1}+N_{2}:n}^{*},\delta_{N_{1}+N_{2}}^{*},R_{N_{1}+N_{2}}^{*}\right)$ by using ${\hat{\lambda }}_{ij}$, $N_{1}$, $N_{2}$, and $\left(R_{1},\cdots ,R_{N_{1}},\cdots ,R_{N_{1}+N_{2}}\right)$, calculate the new MLE of $\lambda_{ij}$, denoted ${\hat{\lambda }}_{ij}^{*[1]}(i,j=1,2)$ from Equation (13).

**Step 3****.** Repeat Step 2 N times so that it is the number of bootstrap samples, and estimators ${\hat{\lambda }}_{ij}^{*[m]}(i,j=1,2;m=1,\cdots ,N)$ can be obtained.

**Step 4****.** Arrange ${\hat{\lambda }}_{ij}^{*[m]}$ in ascending order to obtain the credible interval of the parameter $\lambda_{ij}$, then ${\hat{\lambda }}_{ij}^{*[1]}<{\hat{\lambda }}_{ij}^{*[2]}<\cdots <{\hat{\lambda }}_{ij}^{*[N]}$, i,j=1,2;m=1,⋯,N.

**Step 5****.** Obtain the two-side $100\left(1-\gamma \right)\%$ Bootstrap-p confidence intervals (BPCIs) for parameters as:(17)$$\left({\hat{\lambda }}_{ij}^{*[N*\gamma /2]},\;{\hat{\lambda }}_{ij}^{*[N*(1-\gamma /2)]}\right)$$

#### 3.2.2. Bootstrap-t Method

Hall (1988) developed Bootstrap-t method, and the Bootstrap-t samples are generated as follows:

**Step 1****.** Same as Bootstrap-p method step 1.

**Step 2****.** Same as Bootstrap-p method step 2.

**Step 3****.** Obtain the bootstrap-t statistic ${\hat{T}}_{ij}^{*}=\frac{{\hat{\lambda }}_{ij}^{*[1]}-{\hat{\lambda }}_{ij}}{\sqrt{Var({\hat{\lambda }}_{ij})}}$ for the parameter $\lambda_{ij}$, and denote ${\hat{T}}_{ij}^{*[1]}(i,j=1,2)$.

**Step 4****.** Repeat Steps 2–3 N times that it is the number of bootstrap samples, the estimators ${\hat{T}}_{ij}^{*[m]}(i,j=1,2;m=1,\cdots ,N)$ can be obtained.

**Step 5****.** Arrange ${\hat{T}}_{ij}^{*[m]}(i,j=1,2;m=1,\cdots ,N)$ in ascending order, and obtain the credible interval of the parameter $\lambda_{ij}$, then ${\hat{T}}_{ij}^{*[1]}<{\hat{T}}_{ij}^{*[2]}<\cdots <{\hat{T}}_{ij}^{*[N]}(i,j=1,2)$.

**Step 6****.** Obtain the two-side $100\left(1-\gamma \right)\%$ Bootstrap-t confidence intervals (BTCIs) for parameters as
(18)$$\left({\hat{\lambda }}_{ij}-{\hat{T}}_{ij}^{*[(1-\gamma /2)*N]}\sqrt{Var({\hat{\lambda }}_{ij})},{\hat{\lambda }}_{ij}-{\hat{T}}_{ij}^{*[\gamma /2*N]}\sqrt{Var({\hat{\lambda }}_{ij})}\right)$$

## 4. Bayesian Analysis

In this section, we consider the BEs of $\lambda_{ij}$ based on the squared error loss function (SELF), entropy loss function (ELF), and LINEX (linear-exponential) loss function (LLF). The loss functions expressions are in the Appendix A.

As the conjugate prior, an independent gamma was chosen as prior distribution $Ga(\alpha_{ij},\beta_{ij})$ for $\lambda_{ij}$, so
(19)$$\pi (\lambda_{ij}\left|\alpha_{ij},\beta_{ij}\right.)=\frac{\beta_{ij}{}^{\alpha_{ij}}}{\Gamma (\alpha_{ij})}\lambda_{ij}^{\alpha_{ij}-1}e^{-\beta_{ij}\lambda_{ij}}\propto \lambda_{ij}^{\alpha_{ij}-1}e^{-\beta_{ij}\lambda_{ij}}(\lambda_{ij}\geq 0)$$

Based on Equations (10) and (19), the joint posterior density function of $\lambda_{ij}$ can be written as:(20)$$\pi (\lambda_{11},\lambda_{12},\lambda_{21},\lambda_{22}\left|\xi \right.)=\frac{\prod_{i=1}^{2}\prod_{j=1}^{2}L\left(t\left|\lambda_{ij}\right.\right)\cdot \pi (\lambda_{ij})}{\prod_{i=1}^{2}\prod_{j=1}^{2}\int_{0}^{+\infty }L\left(t\left|\lambda_{ij}\right.\right)\cdot \pi (\lambda_{ij})d\lambda_{ij}}$$
where $\xi =(n_{ij},t_{i},R_{i},\tau ,i=1,\cdots ,N;j=1,2)$.

The marginal posterior distribution of $\lambda_{ij}$ is
(21a)$$\pi (\lambda_{1j}\left|\xi \right.)=\frac{L(t|\lambda_{1j})\cdot \pi (\lambda_{1j})}{\int_{0}^{+\infty }L(t|\lambda_{1j})\cdot \pi (\lambda_{1j})d\lambda_{1j}}=\frac{{\left(\beta_{1j}+T_{1}+T_{21}\right)}^{n_{1j}+\alpha_{1j}-1}}{\Gamma (\alpha_{1j}+n_{1j})}\lambda_{1j}^{n_{1j}+\alpha_{1j}-1}e^{-\lambda_{1j}(\beta_{1j}+T_{1}+T_{21})}\;\;\propto \lambda_{1j}{}^{n_{1j}+\alpha_{1j}-1}e^{-\lambda_{1j}(\beta_{1j}+T_{1}+T_{21})}$$
(21b)$$\pi (\lambda_{2j}\left|\xi \right.)=\frac{L(t|\lambda_{2j})\pi (\lambda_{2j})}{\int_{0}^{+\infty }L(t|\lambda_{2j})\pi (\lambda_{2j})d\lambda_{2j}}=\frac{{\left(\beta_{2j}+T_{22}\right)}^{n_{2j}+\alpha_{2j}-1}}{\Gamma (\alpha_{2j}+n_{2j})}\lambda_{2j}^{n_{2j}+\alpha_{2j}-1}e^{-\lambda_{2j}\left(\beta_{2j}+T_{22}\right)}\;\;\propto \lambda_{2j}{}^{n_{2j}+\alpha_{2j}-1}e^{-\lambda_{2j}(\beta_{2j}+T_{22})}$$

Given the data, the posterior distribution of $\lambda_{1j}$ is $Ga(n_{1j}+\alpha_{1j},\beta_{1j}+T_{1}+T_{21})$, and the posterior distribution of $\lambda_{2j}$ is $Ga(n_{2j}+\alpha_{2j},\beta_{2j}+T_{22}),j=1,2$.

### 4.1. Bayesian Estimation of $\lambda_{ij}$ under SELF

The BE of $\lambda_{ij}$ under the SELF is the expectation of the posterior distribution. Therefore, ${\hat{\lambda }}_{ij(BS)}$ can be derived as
(22)$${\hat{\lambda }}_{ij(BS)}=E\left(\lambda_{ij}\left|\xi \right.\right)=\int_{\Theta }\lambda_{ij}\pi (\lambda_{ij}\left|\xi \right.)d\lambda_{ij},i,j=1,2$$ Thus, the BEs of $\lambda_{1j}$ and $\lambda_{2j}$ are obtained under Equations (21) and (22) as
(23a)$${\hat{\lambda }}_{1j(BS)}=\frac{n_{1j}+\alpha_{1j}}{\beta_{1j}+T_{1}+T_{21}},j=1,2$$
(23b)$${\hat{\lambda }}_{2j(BS)}=\frac{n_{2j}+\alpha_{2j}}{\beta_{2j}+T_{22}},j=1,2$$

### 4.2. Bayesian Estimation of $\lambda_{ij}$ under ELF

The BEs of $\lambda_{ij}(i,j=1,\left.2\right)$ as follows:(24)$${\hat{\lambda }}_{ij(BE)}={\left[E(\lambda_{ij}{}^{-1}|\xi )\right]}^{-1}=1/\int_{\Theta }\lambda_{ij}{}^{-1}\pi (\lambda_{ij}\left|\xi \right.)d\lambda_{ij}$$ Thus, the BEs of $\lambda_{1j}$ and $\lambda_{2j}$ are obtained under Equations (21) and (24) as
(25a)$${\hat{\lambda }}_{1j(BE)}=\frac{n_{1j}+\alpha_{1j}-1}{\beta_{1j}+T_{1}+T_{21}},j=1,2$$
(25b)$${\hat{\lambda }}_{2j(BE)}=\frac{n_{2j}+\alpha_{2j}-1}{\beta_{2j}+T_{22}},j=1,2$$

### 4.3. Bayesian Estimation of $\lambda_{ij}$ under LLF

The BEs of $\lambda_{ij}(i,j=1,\left.2\right)$ as follows:(26)$${\hat{\lambda }}_{ij(BL)}=-\frac{1}{k}\ln[E(e^{-k\lambda_{ij}}\left|\xi \right.)]$$

Thus, the BEs of $\lambda_{1j}$ and $\lambda_{2j}$ are obtained under Equations (21) and (26) as
(27a)$${\hat{\lambda }}_{1j(BL)}=\frac{n_{1j}+\alpha_{1j}-1}{k}\ln(1+\frac{k}{\beta_{1j}+T_{1}+T_{21}}),j=1,2$$
(27b)$${\hat{\lambda }}_{2j(BL)}=\frac{n_{2j}+\alpha_{2j}-1}{k}\ln(1+\frac{k}{\beta_{2j}+T_{22}}),j=1,2$$

## 5. Expected Bayesian Analysis

Since the values of the hyper-parameters are not easy to determine, it has certain randomness. The Expected Bayesian estimation is the average of Bayes estimates of θ with hyper-parameters a and b in the domain Θ. In this section, we will obtain the E-BE of based on the SELF, ELF, and LLF. The definition of E-BE was originally addressed by Han [29] as follows.

**Definition** **1.***With*${\hat{\theta }}_{B}(a,b)$*being continuous,*(28)$${\hat{\theta }}_{EB}=E[{\hat{\theta }}_{B}(a,b)]=\iint_{\Theta }{\hat{\theta }}_{B}(a,b)\pi (a,b)dadb$$*is called the E-BE of*θ*, which is assumed to be finite, where*Θ*is the domain of*a*and*b*,*${\hat{\theta }}_{B}(a,b)$*is the BE of*θ*with hyper-parameters*a*and*b*, and*$\pi \left(a,b\right)$*is the joint density function of*a*and*b*over*Θ*. By Definition 1, the expected Bayesian is the expectation of*${\hat{\theta }}_{B}(a,b)$*with hyper-parameters*a*and*b.

According to the prior information, the $\lambda_{ij}$ is large with a low probability, and the $\lambda_{ij}$ is small with a high probability, so the hyper-parameters $\alpha_{ij}$ and $\beta_{ij}$ should be chosen to guarantee that $\pi (\lambda_{ij}|\alpha_{ij},\beta_{ij})$ is a decreasing function of $\lambda_{ij}$. For more details, see Han [29]. The derivative of $\pi (\lambda_{ij}|\alpha_{ij},\beta_{ij})$ with respect to $\lambda_{ij}$ is,
(29)$$\frac{d\pi (\lambda_{ij}|\alpha_{ij},\beta_{ij})}{d\lambda_{ij}}=\frac{\beta_{ij}{}^{\alpha_{ij}}}{\Gamma (\alpha_{ij})}\lambda_{ij}^{\alpha_{ij}-2}e^{-\beta_{ij}\lambda_{ij}}[(\alpha_{ij}-1)-\beta_{ij}\lambda ]$$ When $0<\alpha_{ij}<1$, $\beta_{ij}>0$, then $\frac{d\pi (\lambda_{ij}|\alpha_{ij},\beta_{ij})}{d\lambda_{ij}}<0$. Given $0<\alpha_{ij}<1$, then the larger the value of $\beta_{ij}$, the thinner the tail of the density function. Berger [33] showed that the hyper parameter $\beta_{ij}$ should be chosen under the restriction $0<\beta_{ij}<c$ (c is a constant). Suppose that $\alpha_{ij}$ and $\beta_{ij}$ are independent, the joint density of $\alpha_{ij}$ and $\beta_{ij}$ is given by $\pi \left(\alpha_{ij},\beta_{ij}\right)=\pi \left(\alpha_{ij}\right)\pi \left(\beta_{ij}\right)$. Depending on different distribution of the parameters $\alpha_{ij}$ and $\beta_{ij}$, E-BE estimation of $\lambda_{ij}$ is obtained. Several authors have applied the E-BE method to analyze data, such as Abdul-Sathar et al. [34] and Shahram [35].

E-BE of $\lambda_{ij}$ is obtained relying on different distributions of $\alpha_{ij}$ and $\beta_{ij}$. These distributions are used to describe the effect of the different prior distributions on E-BE of $\lambda_{ij}$. In this paper, the $\pi \left(\alpha_{ij},\beta_{ij}\right)$ may be used:(30a)$$\pi \left(\alpha_{ij},\beta_{ij}\right)=\frac{1}{c}\;(0<\alpha_{ij}<1,0<\beta_{ij}<c)$$
(30b)$$\pi \left(\alpha_{ij},\beta_{ij}\right)=\frac{2\beta_{ij}}{c^{2}}(0<\alpha_{ij}<1,0<\beta_{ij}<c)$$

### 5.1. E-Bayesian Estimation of $\lambda_{ij}$ under SELF

The E-BEs of $\lambda_{ij}$ are obtained under Equations (23) and (30a) as
(31a)$${\hat{\lambda }}_{1j(EBS1)}=\frac{1}{c}\int_{0}^{1}\int_{0}^{c}\frac{n_{1j}+\alpha_{1j}}{\beta_{1j}+T_{1}+T_{21}}d\alpha_{1j}d\beta_{1j}=\frac{2n_{1j}+1}{2c}\cdot \ln(1+\frac{c}{T_{1}+T_{21}})$$
(31b)$${\hat{\lambda }}_{2j(EBS1)}\frac{1}{c}\int_{0}^{1}\int_{0}^{c}\frac{n_{2j}+\alpha_{2j}}{\beta_{2j}+T_{22}}d\alpha_{2j}d\beta_{2j}=\frac{2n_{2j}+1}{2c}\cdot \ln(1+\frac{c}{T_{22}})$$ The E-BEs of $\lambda_{ij}$ are obtained under Equations (23) and (30b) as
(32a)$${\hat{\lambda }}_{1j(EBS2)}\frac{2}{c^{2}}\int_{0}^{1}\int_{0}^{c}\frac{(n_{1j}+\alpha_{1j})\beta_{1j}}{\beta_{1j}+T_{1}+T_{21}}d\alpha_{1j}d\beta_{1j}=\frac{2n_{1j}+1}{c^{2}}\cdot \left[c-\left(T_{1}+T_{21}\right)\ln(1+\frac{c}{T_{1}+T_{21}})\right]$$
(32b)$${\hat{\lambda }}_{2j(EBS2)}\frac{2}{c^{2}}\int_{0}^{1}\int_{0}^{c}\frac{(n_{2j}+\alpha_{2j})\beta_{2j}}{\beta_{2j}+T_{22}}d\alpha_{2j}d\beta_{2j}=\frac{2n_{2j}+1}{c^{2}}\cdot \left[c-T_{22}\ln(1+\frac{c}{T_{22}})\right]$$

### 5.2. E-Bayesian Estimation of $\lambda_{ij}$ under ELF

The E-BEs of $\lambda_{ij}$ are obtained under Equations (25) and (30a) as
(33a)$${\hat{\lambda }}_{1j(EBE1)}\frac{1}{c}\int_{0}^{1}\int_{0}^{c}\frac{n_{1j}+\alpha_{1j}-1}{\beta_{1j}+T_{1}+T_{21}}d\alpha_{1j}d\beta_{1j}=\frac{2n_{1j}-1}{2c}\cdot \ln(1+\frac{c}{T_{1}+T_{21}})$$
(33b)$${\hat{\lambda }}_{2j(EBE1)}=\frac{1}{c}\int_{0}^{1}\int_{0}^{c}\frac{n_{2j}+\alpha_{2j}-1}{\beta_{2j}+T_{22}}d\alpha_{2j}d\beta_{2j}=\frac{2n_{2j}-1}{2c}\cdot \ln(1+\frac{c}{T_{22}})$$

The E-BEs of $\lambda_{ij}$ are obtained under Equations (25) and (30b) as
(34a)$${\hat{\lambda }}_{1j(EBE2)}\frac{2}{c^{2}}\int_{0}^{1}\int_{0}^{c}\frac{(n_{1j}+\alpha_{1j}-1)\beta_{1j}}{\beta_{1j}+T_{1}+T_{21}}d\alpha_{1j}d\beta_{1j}=\frac{2n_{1j}-1}{c^{2}}\cdot \left[c-\left(T_{1}+T_{21}\right)\ln(1+\frac{c}{T_{1}+T_{21}})\right]$$
(34b)$${\hat{\lambda }}_{3j(EBE3)}\frac{2}{c^{2}}\int_{0}^{1}\int_{0}^{c}\frac{(n_{2j}+\alpha_{2j}-1)\beta_{2j}}{\beta_{2j}+T_{22}}d\alpha_{2j}d\beta_{2j}=\frac{2n_{2j}-1}{c^{2}}\left[c-T_{22}\ln(1+\frac{c}{T_{22}})\right]$$

### 5.3. E-Bayesian Estimation of $\lambda_{ij}$ under LLF

The E-BEs of $\lambda_{ij}$ are obtained under Equations (27) and (30a) as
(35a)$${\hat{\lambda }}_{1j(EBL1)}\frac{1}{c}\int_{0}^{1}\int_{0}^{c}\frac{n_{1j}+\alpha_{1j}-1}{k}\ln(1+\frac{k}{\beta_{1j}+T_{1}+T_{21}})d\alpha_{1j}d\beta_{1j}\;=\frac{2n_{1j}-1}{2k}\left[\ln(1+\frac{k}{c+T_{1}+T_{21}})+\frac{T_{1}+T_{21}+k}{c}\ln(1+\frac{c}{T_{1}+T_{21}+k})-\frac{T_{1}+T_{21}}{c}\ln(1+\frac{c}{T_{1}+T_{21}})\right]$$
(35b)$${\hat{\lambda }}_{2j(EBL2)}\frac{1}{c}\int_{0}^{1}\int_{0}^{c}\frac{n_{2j}+\alpha_{2j}-1}{k}\ln(1+\frac{k}{\beta_{2j}+T_{22}})d\alpha_{2j}d\beta_{2j}\;=\frac{2n_{2j}-1}{2k}\left[\ln(1+\frac{k}{c+T_{22}})+\frac{T_{22}+k}{c}\ln(1+\frac{c}{T_{22}+k})-\frac{T_{22}}{c}\ln(1+\frac{c}{T_{22}})\right]$$

The E-BEs of $\lambda_{ij}$ are obtained under Equations (27) and (30b) as
(36a)$${\hat{\lambda }}_{1j(EBL2)}\frac{2}{c^{2}}\int_{0}^{1}\int_{0}^{c}\frac{n_{1j}+\alpha_{1j}-1}{k}\beta_{1j}\ln(1+\frac{k}{\beta_{1j}+T_{1}+T_{21}})d\alpha_{1j}d\beta_{1j}\;=\frac{2n_{1j}-1}{2k}\left[\ln(1+\frac{k}{c+T_{1}+T_{21}})-\frac{{\left(T_{1}+T_{21}+k\right)}^{2}}{c^{2}}\ln(1+\frac{c}{T_{1}+T_{21}+k})+\frac{{\left(T_{1}+T_{21}\right)}^{2}}{c^{2}}\ln(1+\frac{c}{T_{1}+T_{21}})+\frac{k}{c}\right]$$
(36b)$${\hat{\lambda }}_{2j(EBL2)}\frac{2}{c^{2}}\int_{0}^{1}\int_{0}^{c}\frac{n_{2j}+\alpha_{2j}-1}{k}\beta_{2j}\ln(1+\frac{k}{\beta_{2j}+T_{22}})d\alpha_{2j}d\beta_{2j}\;=\frac{2n_{2j}-1}{2k}\left\{\ln\left(1+\frac{k}{c+T_{22}}\right)-\frac{{\left(T_{22}+k\right)}^{2}}{c^{2}}\ln\left(1+\frac{c}{T_{22}+k}\right)+\frac{T_{22}{}^{2}}{c^{2}}\ln\left(1+\frac{c}{T_{22}}\right)+\frac{k}{c}\right\}$$

## 6. Hierarchical Bayesian Estimation

In this section, we obtain the H-BEs of $\lambda_{ij}$ based on the SELF, ELF, and LLF. The definition of H-BE was originally addressed by Lindley and Smith [28] as follows.

**Definition** **2.***If*a and b*are hyper-parameters in the parameter of*θ*, the prior density function of*θ*is*π(θ|a,b)*, and the prior density function of the hyper-parameters of*a and b*is*π(a,b)*, then the hierarchical prior (H-P) density function of*θ*is defined as follows:*(37)$$\pi (\theta )=\iint_{\Theta }\pi (\theta |a,b)\pi \left(a,b\right)dadb$$*where*Θ*is the domain of*a and b.

The H-P density functions of the parameters $\lambda_{ij}$ are obtained under Equations (19) and (30a) as:(38)$$\pi (\lambda_{ij})=\frac{1}{c}\int_{0}^{1}\int_{0}^{c}\frac{\beta_{ij}^{\alpha_{ij}}}{\Gamma (\alpha_{ij})}\lambda_{ij}^{\alpha_{ij}-1}e^{-\beta_{ij}\lambda_{ij}}d\alpha_{ij}d\beta_{ij}$$

We get the posterior density function of $\lambda_{ij}$ under Equations (20) and (38) as:(39a)$$\begin{array}{cc} \pi (\lambda_{1j}|\xi )= & \frac{\pi (t|\lambda_{1j})\pi (\lambda_{1j})}{\int_{0}^{+\infty }\pi (t|\lambda_{1j})\pi (\lambda_{1j})d\lambda_{1j}}\\ & \;=\frac{\lambda_{1j}^{n_{1j}}\times e^{-\lambda_{1j}\left(T_{1}+T_{21}\right)}\times \frac{1}{c}\int_{0}^{1}\int_{0}^{c}\frac{\beta_{1j}^{\alpha_{1j}}}{\Gamma (\alpha_{1j})}\lambda_{1j}^{\alpha_{1j}-1}e^{-\beta_{1j}\lambda_{1j}}d\alpha_{1j}d\beta_{1j}}{\frac{1}{c}\int_{0}^{1}\int_{0}^{c}\frac{\beta_{1j}^{\alpha_{1j}}}{\Gamma (\alpha_{1j})}(\int_{0}^{+\infty }\lambda_{1j}^{n_{1j}+\alpha_{1j}-1}e^{-(\beta_{1j}+T_{1}+T_{21})\lambda_{1j}}d\lambda_{1j})d\alpha_{1j}d\beta_{1j}}\\ & \;=\frac{\lambda_{1j}^{n_{1j}}\times e^{-\lambda_{1j}\left(T_{1}+T_{21}\right)}\times \int_{0}^{1}\int_{0}^{c}\frac{\beta_{1j}{}^{\alpha_{1j}}}{\Gamma (\alpha_{1j})}\lambda_{1j}^{\alpha_{1j}-1}e^{-\beta_{1j}\lambda_{1j}}d\alpha_{1j}d\beta_{1j}}{\int_{0}^{1}\int_{0}^{c}\frac{\Gamma (\alpha_{1j}+n_{1j})\beta_{1j}{}^{\alpha_{1j}}}{\Gamma (\alpha_{1j}){\left(\beta_{1j}+T_{1}+T_{21}\right)}^{n_{1j}+\alpha_{1j}-1}}d\alpha_{1j}d\beta_{1j}} \end{array}$$ In the same way,
(39b)$$\pi (\lambda_{2j}\left|\xi \right.)=\frac{\lambda_{2j}^{n_{2j}}\times e^{-\lambda_{2j}T_{22}}\times \int_{0}^{1}\int_{0}^{c}\frac{\beta_{2j}^{\alpha_{2j}}}{\Gamma (\alpha_{2j})}\lambda_{2j}^{\alpha_{2j}-1}e^{-\beta_{2j}\lambda_{2j}}d\alpha_{2j}d\beta_{2j}}{\int_{0}^{1}\int_{0}^{c}\frac{\Gamma (\alpha_{2j}+n_{2j})\beta_{2j}^{\alpha_{2j}}}{\Gamma (\alpha_{2j}){\left(\beta_{2j}+T_{22}\right)}^{n_{2j}+\alpha_{2j}-1}}d\alpha_{2j}d\beta_{2j}}$$

The H-P density function of the parameters $\lambda_{ij}$ are obtained under Equations (19) and (30b) as:(40)$$\pi (\lambda_{ij})=\frac{2}{c^{2}}\int_{0}^{1}\int_{0}^{c}\frac{\beta_{ij}{}^{\alpha_{ij}+1}}{\Gamma (\alpha_{ij})}\lambda_{ij}^{\alpha_{ij}-1}e^{-\beta_{ij}\lambda_{ij}}d\alpha_{ij}d\beta_{ij}$$

We get the posterior density function of $\lambda_{ij}$ under Equations (20) and (40) as:(41a)$$\pi (\lambda_{1j}|\xi )=\frac{\lambda_{1j}^{n_{1j}}\times e^{-\lambda_{1j}(T_{1}+T_{21})}\times \int_{0}^{1}\int_{0}^{c}\frac{\beta_{1j}^{\alpha_{1j}+1}}{\Gamma (\alpha_{1j})}\lambda_{1j}^{\alpha_{1j}-1}e^{-\beta_{1j}\lambda_{1j}}d\alpha_{1j}d\beta_{1j}}{\int_{0}^{1}\int_{0}^{c}\frac{\Gamma (\alpha_{1j}+n_{1j})\beta_{1j}{}^{\alpha_{1j}+1}}{\Gamma (\alpha_{1j}){\left(\beta_{1j}+T_{1}+T_{21}\right)}^{n_{1j}+\alpha_{1j}-1}}d\alpha_{1j}d\beta_{1j}}$$
(41b)$$\pi (\lambda_{2j}\left|\xi \right.)=\frac{\lambda_{2j}^{n_{2j}}\times e^{-\lambda_{2j}T_{22}}\times \int_{0}^{1}\int_{0}^{c}\frac{\beta_{2j}^{\alpha_{2j}+1}}{\Gamma (\alpha_{2j})}\lambda_{2j}^{\alpha_{2j}-1}e^{-\beta_{2j}\lambda_{2j}}d\alpha_{2j}d\beta_{2j}}{\int_{0}^{1}\int_{0}^{c}\frac{\Gamma (\alpha_{2j}+n_{2j})\beta_{2j}^{\alpha_{2j}+1}}{\Gamma (\alpha_{2j}){\left(\beta_{2j}+T_{22}\right)}^{n_{2j}+\alpha_{2j}-1}}d\alpha_{2j}d\beta_{2j}}$$

### 6.1. H-Bayesian Estimation of $\lambda_{ij}$ under SELF

The H-BEs of $\lambda_{ij}$ are obtained under Equations (22) and (39) as:(42a)$$\begin{array}{cc} {\hat{\lambda }}_{1j(HBS1)} & =\int_{0}^{+\infty }\lambda_{1j}\pi (\lambda_{1j}\left|\xi \right.)d\lambda_{1j}\\ & \;\;\;\;\;=\frac{\int_{0}^{1}\int_{0}^{c}\frac{\Gamma (\alpha_{1j}+n_{1j}+1)\beta_{1j}^{\alpha_{1j}}}{\Gamma (\alpha_{1j}){\left(\beta_{1j}+T_{1}+T_{21}\right)}^{n_{1j}+\alpha_{1j}}}d\alpha_{1j}d\beta_{1j}}{\int_{0}^{1}\int_{0}^{c}\frac{\Gamma (\alpha_{1j}+n_{1j})\beta_{1j}^{\alpha_{1j}}}{\Gamma (\alpha_{1j}){\left(\beta_{1j}+T_{1}+T_{21}\right)}^{n_{1j}+\alpha_{1j}-1}}d\alpha_{1j}d\beta_{1j}} \end{array}$$
(42b)$$\begin{array}{cc} {\hat{\lambda }}_{2j(HBS1)} & =\int_{0}^{+\infty }\lambda_{2j}\pi (\lambda_{2j}\left|\xi \right.)d\lambda_{2j}\\ & \;\;\;\;\;=\frac{\int_{0}^{1}\int_{0}^{c}\frac{\Gamma (\alpha_{2j}+n_{2j}+1)\beta_{2j}^{\alpha_{2j}}}{\Gamma (\alpha_{2j}){\left(\beta_{1j}+T_{22}\right)}^{n_{2j}+\alpha_{2j}}}d\alpha_{2j}d\beta_{2j}}{\int_{0}^{1}\int_{0}^{c}\frac{\Gamma (\alpha_{2j}+n_{2j})\beta_{2j}^{\alpha_{2j}}}{\Gamma (\alpha_{2j}){\left(\beta_{2j}+T_{22}\right)}^{n_{2j}+\alpha_{2j}-1}}d\alpha_{2j}d\beta_{2j}} \end{array}$$

The H-BEs of $\lambda_{ij}$ are obtained under Equations (22) and (41) as:(43a)$$\begin{array}{cc} {\hat{\lambda }}_{1j(HBS2)} & =\int_{0}^{+\infty }\lambda_{1j}\pi (\lambda_{1j}\left|\xi \right.)d\lambda_{1j}\\ & \;\;\;\;\;=\frac{\int_{0}^{1}\int_{0}^{c}\frac{\Gamma (\alpha_{1j}+n_{1j}+1)\beta_{1j}^{\alpha_{1j}+1}}{\Gamma (\alpha_{1j}){\left(\beta_{1j}+T_{1}+T_{21}\right)}^{n_{1j}+\alpha_{1j}}}d\alpha_{1j}d\beta_{1j}}{\int_{0}^{1}\int_{0}^{c}\frac{\Gamma (\alpha_{1j}+n_{1j})\beta_{1j}^{\alpha_{1j}+1}}{\Gamma (\alpha_{1j}){\left(\beta_{1j}+T_{1}+T_{21}\right)}^{n_{1j}+\alpha_{1j}-1}}d\alpha_{1j}d\beta_{1j}} \end{array}$$
(43b)$$\begin{array}{cc} {\hat{\lambda }}_{2j(HBS2)} & =\int_{0}^{+\infty }\lambda_{2j}\pi (\lambda_{2j}\left|\xi \right.)d\lambda_{2j}\\ & \;\;\;\;\;=\frac{\int_{0}^{1}\int_{0}^{c}\frac{\Gamma (\alpha_{2j}+n_{2j}+1)\beta_{2j}^{\alpha_{2j}+1}}{\Gamma (\alpha_{2j}){\left(\beta_{2j}+T_{22}\right)}^{n_{2j}+\alpha_{2j}}}d\alpha_{2j}d\beta_{2j}}{\int_{0}^{1}\int_{0}^{c}\frac{\Gamma (\alpha_{2j}+n_{2j})\beta_{2j}^{\alpha_{2j}+1}}{\Gamma (\alpha_{2j}){\left(\beta_{2j}+T_{22}\right)}^{n_{2j}+\alpha_{2j}-1}}d\alpha_{2j}d\beta_{2j}} \end{array}$$

### 6.2. H-Bayesian Estimation of $\lambda_{ij}$ under ELF

The H-BEs of $\lambda_{ij}$ are obtained under Equations (24) and (39) as:(44a)$$\begin{array}{cc} {\hat{\lambda }}_{1j(HBE1)} & =1/\int_{0}^{+\infty }\lambda_{1j}^{-1}\pi (\lambda_{1j}\left|\xi \right.)d\lambda_{1j}\\ & \;\;\;\;\;=\frac{\int_{0}^{1}\int_{0}^{c}\frac{\Gamma (\alpha_{1j}+n_{1j})\beta_{1j}^{\alpha_{1j}}}{\Gamma (\alpha_{1j}){\left(\beta_{1j}+T_{1}+T_{21}\right)}^{n_{1j}+\alpha_{1j}-1}}d\alpha_{1j}d\beta_{1j}}{\int_{0}^{1}\int_{0}^{c}\frac{\Gamma (\alpha_{1j}+n_{1j}-1)\beta_{1j}^{\alpha_{1j}}}{\Gamma (\alpha_{1j}){\left(\beta_{1j}+T_{1}+T_{21}\right)}^{n_{1j}+\alpha_{1j}-2}}d\alpha_{1j}d\beta_{1j}} \end{array}$$
(44b)$$\begin{array}{cc} {\hat{\lambda }}_{2j(HBE1)}= & 1/\int_{0}^{+\infty }\lambda_{2j}^{-1}\pi (\lambda_{2j}\left|\xi \right.)d\lambda_{2j}\\ & \;\;\;\;\;=\frac{\int_{0}^{1}\int_{0}^{c}\frac{\Gamma (\alpha_{2j}+n_{2j})\beta_{2j}^{\alpha_{2j}}}{\Gamma (\alpha_{2j}){\left(\beta_{2j}+T_{22}\right)}^{n_{2j}+\alpha_{2j}-1}}d\alpha_{2j}d\beta_{2j}}{\int_{0}^{1}\int_{0}^{c}\frac{\Gamma (\alpha_{2j}+n_{2j}-1)\beta_{2j}^{\alpha_{2j}}}{\Gamma (\alpha_{2j}){\left(\beta_{2j}+T_{22}\right)}^{n_{2j}+\alpha_{2j}-2}}d\alpha_{2j}d\beta_{2j}} \end{array}$$

The H-BEs of $\lambda_{ij}$ are obtained under Equations (24) and (41) as:(45a)$$\begin{array}{cc} {\hat{\lambda }}_{1j(HBE2)}= & 1/\int_{0}^{+\infty }\lambda_{1j}^{-1}\pi (\lambda_{1j}\left|\xi \right.)d\lambda_{1j}\\ & \;\;\;\;\;=\frac{\int_{0}^{1}\int_{0}^{c}\frac{\Gamma (\alpha_{1j}+n_{1j})\beta_{1j}{}^{\alpha_{1j}+1}}{\Gamma (\alpha_{1j}){\left(\beta_{1j}+T_{1}+T_{21}\right)}^{n_{1j}+\alpha_{1j}-1}}d\alpha_{1j}d\beta_{1j}}{\int_{0}^{1}\int_{0}^{c}\frac{\Gamma (\alpha_{1j}+n_{1j}-1)\beta_{1j}{}^{\alpha_{1j}+1}}{\Gamma (\alpha_{1j}){\left(\beta_{1j}+T_{1}+T_{21}\right)}^{n_{1j}+\alpha_{1j}-2}}d\alpha_{1j}d\beta_{1j}} \end{array}$$
(45b)$$\begin{array}{cc} {\hat{\lambda }}_{2j(HBE2)}= & 1/\int_{0}^{+\infty }\lambda_{2j}^{-1}\pi (\lambda_{2j}\left|\xi \right.)d\lambda_{2j}\\ & \;\;\;\;\;=\frac{\int_{0}^{1}\int_{0}^{c}\frac{\Gamma (\alpha_{2j}+n_{2j})\beta_{2j}{}^{\alpha_{2j}+1}}{\Gamma (\alpha_{2j}){\left(\beta_{2j}+T_{22}\right)}^{n_{2j}+\alpha_{2j}-1}}d\alpha_{2j}d\beta_{2j}}{\int_{0}^{1}\int_{0}^{c}\frac{\Gamma (\alpha_{2j}+n_{2j}-1)\beta_{2j}{}^{\alpha_{2j}+1}}{\Gamma (\alpha_{2j}){\left(\beta_{2j}+T_{22}\right)}^{n_{2j}+\alpha_{2j}-2}}d\alpha_{2j}d\beta_{2j}} \end{array}$$

### 6.3. H-Bayesian Estimation of $\lambda_{ij}$ under LLF

The H-BEs of $\lambda_{ij}$ are obtained under Equations (26) and (39) as:(46a)$$\begin{array}{cc} {\hat{\lambda }}_{1j(HBL1)}= & -\frac{1}{k}\ln\left(\int_{0}^{+\infty }e^{-k\lambda_{1j}}\pi (\lambda_{1j}\left|\xi \right.)d\lambda_{1j}\right)\\ & \;\;=-\frac{1}{k}\ln\frac{\int_{0}^{1}\int_{0}^{c}\frac{\Gamma (\alpha_{1j}+n_{1j})\beta_{1j}^{\alpha_{1j}}}{\Gamma (\alpha_{1j}){\left(\beta_{1j}+T_{1}+T_{21}+k\right)}^{n_{1j}+\alpha_{1j}-1}}d\alpha_{1j}d\beta_{1j}}{\int_{0}^{1}\int_{0}^{c}\frac{\Gamma (\alpha_{1j}+n_{1j})\beta_{1j}^{\alpha_{1j}}}{\Gamma (\alpha_{1j}){\left(\beta_{1j}+T_{1}+T_{21}\right)}^{n_{1j}+\alpha_{1j}-1}}d\alpha_{1j}d\beta_{1j}} \end{array}$$
(46b)$$\begin{array}{cc} {\hat{\lambda }}_{2j(HBL1)}= & -\frac{1}{k}\ln\left(\int_{0}^{+\infty }e^{-k\lambda_{2j}}\pi (\lambda_{2j}|\xi )d\lambda_{2j}\right)\\ & \;\;=-\frac{1}{k}\ln\frac{\int_{0}^{1}\int_{0}^{c}\frac{\Gamma (\alpha_{2j}+n_{2j})\beta_{2j}^{\alpha_{2j}}}{\Gamma (\alpha_{2j}){\left(\beta_{2j}+T_{22}+k\right)}^{n_{2j}+\alpha_{2j}-1}}d\alpha_{2j}d\beta_{2j}}{\int_{0}^{1}\int_{0}^{c}\frac{\Gamma (\alpha_{2j}+n_{2j})\beta_{2j}^{\alpha_{2j}}}{\Gamma (\alpha_{2j}){\left(\beta_{2j}+T_{22}\right)}^{n_{2j}+\alpha_{2j}-1}}d\alpha_{2j}d\beta_{2j}} \end{array}$$

The H-BEs of $\lambda_{ij}$ are obtained under Equations (26) and (41) as:(47a)$$\begin{array}{cc} {\hat{\lambda }}_{1j(HBL2)}= & -\frac{1}{k}\ln\left(\int_{0}^{+\infty }e^{-k\lambda_{1j}}\pi (\lambda_{1j}|\xi )d\lambda_{1j}\right)\\ & \;\;=-\frac{1}{k}\ln\frac{\int_{0}^{1}\int_{0}^{c}\frac{\Gamma (\alpha_{1j}+n_{1j})\beta_{1j}^{\alpha_{1j}+1}}{\Gamma (\alpha_{1j}){\left(\beta_{1j}+T_{1}+T_{21}+k\right)}^{n_{1j}+\alpha_{1j}-1}}d\alpha_{1j}d\beta_{1j}}{\int_{0}^{1}\int_{0}^{c}\frac{\Gamma (\alpha_{1j}+n_{1j})\beta_{1j}^{\alpha_{1j}+1}}{\Gamma (\alpha_{1j}){\left(\beta_{1j}+T_{1}+T_{21}\right)}^{n_{1j}+\alpha_{1j}-1}}d\alpha_{1j}d\beta_{1j}} \end{array}$$
(47b)$$\begin{array}{cc} {\hat{\lambda }}_{2j(HBL2)}= & -\frac{1}{k}\ln\left(\int_{0}^{+\infty }e^{-k\lambda_{2j}}\pi (\lambda_{2j}|\xi )d\lambda_{2j}\right)\\ & \;\;=-\frac{1}{k}\ln\frac{\int_{0}^{1}\int_{0}^{c}\frac{\Gamma (\alpha_{2j}+n_{2j})\beta_{2j}^{\alpha_{2j}+1}}{\Gamma (\alpha_{2j}){\left(\beta_{2j}+T_{22}+k\right)}^{n_{2j}+\alpha_{2j}-1}}d\alpha_{2j}d\beta_{2j}}{\int_{0}^{1}\int_{0}^{c}\frac{\Gamma (\alpha_{2j}+n_{2j})\beta_{2j}^{\alpha_{2j}+1}}{\Gamma (\alpha_{2j}){\left(\beta_{2j}+T_{22}\right)}^{n_{2j}+\alpha_{2j}-1}}d\alpha_{2j}d\beta_{2j}} \end{array}$$

### 6.4. Highest Posterior Density (HPD) Credible Intervals (CRIs)

Here, we propose the following algorithm to compute the associated CRI and HPD CRI (reference [36]).

**Step 1****.** Set N=1000, and generate a Markov Chain Monte Carlo sample $\left({\hat{\lambda }}_{ijk}^{*},i,j=1,2;k=1,\cdots ,N\right)$ from $\pi (\lambda_{ij}|\xi )$.

**Step 2****.** Arrange the sample in ascending order to obtain the CRIs of the parameters $\lambda_{ij}$, then ${\hat{\lambda }}_{ij}^{*[1]}<{\hat{\lambda }}_{ij}^{*[2]}<\cdots <{\hat{\lambda }}_{ij}^{*[N]},i,j=1,2$.

**Step 3****.** Obtain the two-side $100\left(1-\gamma \right)\%$ CRIs for parameters as
(48)$$\left({\hat{\lambda }}_{ij}^{*[N\gamma /2]},{\hat{\lambda }}_{ij}^{*[N(1-\gamma /2)]}\right)$$

**Step 4****.** To construct $100\left(1-\gamma \right)\%$ HPDCRIs of the parameter $\lambda_{ij}$, consider the set of CRIs $\left({\hat{\lambda }}_{ij}^{*[m]},{\hat{\lambda }}_{ij}^{*[m+(1-\gamma )N]}\right),m=1,2,\cdots [\gamma N]$. The $100\left(1-\gamma \right)\%$ HPDCRIs of the parameters $\lambda_{ij}$ is $\left({\hat{\lambda }}_{ij}^{*[m^{*}]},{\hat{\lambda }}_{ij}^{*[m^{*}+(1-\gamma )N]}\right)$, where $m^{*}$ is such that
(49)$${\hat{\lambda }}_{ij}^{*[m^{*}+(1-\gamma )N]}-{\hat{\lambda }}_{ij}^{*[m^{*}]}<{\hat{\lambda }}_{ij}^{*[m+(1-\gamma )N]}-{\hat{\lambda }}_{ij}^{*[m]}$$
for all m=1,2,⋯[γN].

Therefore, the $100\left(1-\gamma \right)\%$ HPDCRIs have the smallest interval width from all the CRIs found.

## 7. Simulation Study and Data Analysis

### 7.1. Simulation Study

In order to investigate the proposed methods, we use Monte Carlo simulations to compare different methods for different sample size and different progressive censoring schemes, which are shown in Table 1. The values of the parameters are chosen to be $\lambda_{11}=2.0$, $\lambda_{12}=1.0$, $\lambda_{21}=4.0$, $\lambda_{22}=2.0$, $N^{*}=1000$, c=1/5 and k=4. For different censoring schemes, we compute the Average Estimates (AEs) and the mean square errors (MSEs) of the MLEs, BEs, E-BEs, and H-BEs, respectively. AEs and MSEs of the estimator of $\lambda_{ij}$ can be calculated as
(50)$$AE_{ij}=\frac{1}{N^{*}}\sum_{k=1}^{N^{*}}{\hat{\lambda }}_{ij}^{\left(k\right)}\;$$
(51)$$MSE_{ij}=\sqrt{\frac{1}{N^{*}}\sum_{k=1}^{N^{*}}{\left(\lambda_{ij}-{\hat{\lambda }}_{ij}^{\left(k\right)}\right)}^{2}}$$
where ${\hat{\lambda }}_{ij}^{\left(k\right)}$ is the kth estimator of the parameter $\lambda_{ij}(i,j=1,2)$.

Simulation study has been done according to the following steps:

**Step 1****.** For given $\lambda_{11}=2.0$, $\lambda_{12}=1.0$, $\lambda_{21}=4.0\lambda_{22}=2.0$, generate a PT-IIC samples based on different sample sizes.

**Step 2****.** The MLEs and the asymptotic CIs of the parameter $\lambda_{ij}(i,j=1,2)$ are computed using Equations (14) and (16).

**Step 3****.** BPCIs and BTCIs are obtained for the parameter $\lambda_{ij}(i,j=1,2)$ using Equations (17) and (18). Here, we have taken B=1000 in Bootstrap CIs.

**Step 4****.** The BEs are computed using Equations (23), (25) and (27), the E-BEs are computed using Equations (31)–(36), and the H-BEs are computed using Equations (42)–(47). Here, the PHD CRIs are also obtained using Equation (49).

**Step 5****.** Repeat step 1 to step 4 $N^{*}$ times and calculate the AEs and MSEs for each estimate. The results are presented in Table 2, Table 3, Table 4, Table 5, Table 6, Table 7, Table 8, Table 9, Table 10, Table 11, Table 12 and Table 13.

**Step 6****.** Compute the average lengths (Als) and the coverage probabilities (CPs) of the 95% asymptotic Cis of the MLEs, the Bootstrap-p method, the Bootstrap-t method, the Bayesian method, the EB method, and the HB method. The results are presented in Table 14, Table 15, Table 16 and Table 17.

**Step 7****.** Compute the Als and the CPs of the HPD CRIs using the Bayesian method, the EB method, and the HB method, and the results are presented in Table 18, Table 19, Table 20 and Table 21.

**Table 1 entropy-24-01405-t001:** The prefixed sample sizes and PT-IIC cases.

Scheme	n	$N_{1}$	$N_{2}$	$\sum_{i=1}^{N_{1}}R_{i}$	$\sum_{i=N_{1}+1}^{N_{2}}R_{i}$	$\left(R_{1},\cdots ,R_{N_{1}})(R_{N_{1}+1},\cdots ,R_{N_{1}+N_{2}}\right)$
1	40	15	15	5	5	(0,…,0,1,1,2,1) (0,…,0,1,1,2,1)
		20	10	5	5	(0,…,0,1,2,2) (0,…,0,1,2,2)
		10	20	5	5	(0,…,0,1,2,2) (0,…,0,1,2,2)
2	60	23	23	7	7	(0,…,0,1,2,2,2) (0,…,0,1,2,2,2)
		30	16	8	6	(0,…,0,2,2,2,2) (0,…,0,2,2,2,2)
		16	30	6	8	(0,…,0,2,2,2) (0,…,0,2,2,2,2)
3	80	30	30	10	10	(1,1,2,1,0,…,1,2,1,1) (1,1,2,1,0,…,1,2,1,1)
		40	20	14	6	(2,2,2,1,0,…,0,1,2,2,2) (1,1,1,0,…,0,1,1,1)
		20	40	6	14	(1,1,1,0,…,0,1,1,1) (2,2,2,1,0,…0,1,2,2,2)

**Table 2 entropy-24-01405-t002:** AEs and MSEs of the parameter $\lambda_{11}=2$ based on SELF.

n	$N_{1}$	$N_{2}$	${\hat{\lambda }}_{11MLE}$		${\hat{\lambda }}_{11BS}$		${\hat{\lambda }}_{11EBS}$		${\hat{\lambda }}_{11HBS}$		Best Estimator
			AE	MSE	AE	MSE	AE	MSE	AE	MSE	
40	15	15	2.1316	0.3675	2.1096	0.2468	2.1184 2.1186	0.2479 0.2393	2.1303 2.1324	0.2127 0.1796	Bayesian
	20	10	1.9062	0.2472	2.0919	0.2085	2.0926 2.0942	0.1945 0.1914	2.1075 2.1123	0.2110 0.2063	E-Bayesian
	10	20	1.8803	0.4243	2.1068	0.3296	1.9524 1.9449	0.2451 0.2557	2.1519 2.1570	0.3470 0.3449	E-Bayesian
60	23	23	2.1254	0.2442	2.0999	0.2549	1.9859 1.9780	0.2123 0.2091	2.1227 2.1152	0.2425 0.2124	E-Bayesian
	30	16	2.0985	0.2049	2.1071	0.2037	1.9273 1.9197	0.1951 0.1901	2.0932 2.0980	0.2031 0.2003	E-Bayesian
	16	30	2.1434	0.3891	2.1207	0.2645	1.8931 1.8860	0.2072 0.2045	2.1113 2.1087	0.1166 0.1019	H-Bayesian
80	30	30	2.1136	0.2258	1.9085	0.2485	1.9216 1.9249	0.2143 0.2112	2.0962 2.1036	0.2040 0.2078	E-Bayesian
	40	20	2.1078	0.1519	2.0905	0.1318	1.9602 1.9630	0.1588 0.1572	2.0461 2.0381	0.1303 0.1322	H-Bayesian
	20	40	2.1399	0.4156	2.1530	0.3687	2.1537 2.1539	0.3251 0.3438	2.0943 2.1067	0.1377 0.1378	H-Bayesian

**Table 3 entropy-24-01405-t003:** AEs and MSEs of the parameter $\lambda_{11}=2$ based on ELF.

n	$N_{1}$	$N_{2}$	${\hat{\lambda }}_{11MLE}$		${\hat{\lambda }}_{11BE}$		${\hat{\lambda }}_{11EBE}$		${\hat{\lambda }}_{11HBE}$		Best Estimator
			AE	MSE	AE	MSE	AE	MSE	AE	MSE	
40	15	15	2.1316	0.3675	2.1096	0.2468	2.0687 2.0592	0.1897 0.1821	2.1296 2.1275	0.2076 0.2080	E-Bayesian
	20	10	1.9062	0.2472	2.0919	0.2085	2.0143 2.0162	0.1382 0.1377	2.0760 2.0786	0.1502 0.1573	E-Bayesian
	10	20	1.8803	0.4243	2.1068	0.3296	1.8916 1.8985	0.3546 0.3549	2.1107 2.1039	0.3380 0.3479	Bayesian
60	23	23	2.1254	0.2442	2.0999	0.2549	1.9082 1.9106	0.2043 0.2013	2.0982 2.0901	0.2124 0.2301	E-Bayesian
	30	16	2.0985	0.2049	2.1071	0.2037	1.9439 1.9366	0.2714 0.2667	2.0956 2.1010	0.1884 0.1879	E-Bayesian
	16	30	2.1434	0.3891	2.1207	0.2645	1.8203 1.8067	0.2962 0.2912	2.1189 2.1027	0.1978 0.1830	H-Bayesian
80	30	30	2.1136	0.2258	1.9085	0.2485	1.9114 1.9149	0.2170 0.2140	2.0937 2.1025	0.2252 0.2267	E-Bayesian
	40	20	2.1078	0.1519	2.0905	0.1318	1.9012 1.8943	0.1528 0.1512	2.0286 2.0417	0.1282 0.1328	H-Bayesian
	20	40	2.1399	0.4156	2.1530	0.3687	2.1560 2.1566	0.3763 0.3592	2.1097 2.1003	0.1128 0.1129	H-Bayesian

**Table 4 entropy-24-01405-t004:** AEs and MSEs of the parameter $\lambda_{11}=2$ based on LLF.

n	$N_{1}$	$N_{2}$	${\hat{\lambda }}_{11MLE}$		${\hat{\lambda }}_{11BL}$		${\hat{\lambda }}_{11EBL}$		${\hat{\lambda }}_{11HBL}$		**Best Estimator**
			AE	MSE	AE	MSE	AE	MSE	AE	MSE	
40	15	15	2.1316	0.3675	2.1096	0.2468	2.0899 2.0806	0.1552 0.1384	1.9033 1.9002	0.2183 0.2048	E-Bayesian
	20	10	1.9062	0.2472	2.0919	0.2085	1.9982 1.9902	0.1373 0.1336	1.9589 1.9596	0.1540 0.1573	E-Bayesian
	10	20	1.8803	0.4243	2.1068	0.3296	1.8974 1.9001	0.4364 0.4402	1.8978 1.8981	0.3942 2.3899	Bayesian
60	23	23	2.1254	0.2442	2.0999	0.2549	1.9231 1.9257	0.2384 0.2354	1.9777 1.9729	0.2005 0.2034	H-Bayesian
	30	16	2.0985	0.2049	2.1071	0.2037	1.9294 1.9223	0.2624 0.2779	1.88115 1.8821	0.2612 0.2681	E-Bayesian
	16	30	2.1434	0.3891	2.1207	0.2645	1.8967 1.8992	0.3912 0.3878	1.8862 1.8879	0.2804 0.2832	Bayesian
80	30	30	2.1136	0.2258	1.9085	0.2485	1.9185 1.9178	0.1911 0.1882	1.9192 1.9224	0.2528 0.2446	E-Bayesian
	40	20	2.1078	0.1519	2.0905	0.1318	1.9043 1.9075	0.1512 0.1496	2.0596 2.0630	0.1308 0.1353	H-Bayesian
	20	40	2.1399	0.4156	2.1530	0.3687	2.1474 2.1574	0.3265 0.3329	1.9133 1.9262	0.2101 0.2214	H-Bayesian

**Table 5 entropy-24-01405-t005:** AEs and MSEs of the parameter $\lambda_{12}=1$ based on SELF.

n	$N_{1}$	$N_{2}$	${\hat{\lambda }}_{12MLE}$		${\hat{\lambda }}_{12BS}$		${\hat{\lambda }}_{12EBS}$		${\hat{\lambda }}_{12HBS}$		Best Estimator
			AE	MSE	AE	MSE	AE	MSE	AE	MSE	
40	15	15	1.1003	0.1838	1.0913	0.1027	1.1193 1.1148	0.1115 0.1106	1.1468 1.1226	0.1372 0.1327	Bayesian
	20	10	1.0576	0.1601	1.0912	0.1186	1.0477 1.0533	0.1084 0.1074	1.1240 1.1219	0.1581 0.1623	E-Bayesian
	10	20	1.1408	0.2634	1.1717	0.2732	0.8999 0.8958	0.1129 0.1058	1.1526 1.1578	0.2228 0.2178	E-Bayesian
60	23	23	1.1217	0.1944	1.1131	0.1578	1.1003 1.1072	0.1486 0.1369	1.1148 1.1141	0.1780 0.1776	E-Bayesian
	30	16	1.0917	0.11457	1.0994	0.1185	1.0848 1.0801	0.1449 0.1433	1.1096 1.1075	0.1348 0.1388	E-Bayesian
	16	30	1.1888	0.2456	1.2075	0.2338	1.1925 1.1883	0.2225 0.2201	1.1736 1.1739	0.2135 0.2269	H-Bayesian
80	30	30	1.1232	0.1987	1.1311	0.1297	1.1482 1.1446	0.1671 0.1658	1.1202 1.1245	0.1138 0.1172	H-Bayesian
	40	20	0.9247	0.1302	1.1080	0.1239	0.9460 0.9524	0.1163 0.1148	1.0239 1.0268	0.1128 0.1078	H-Bayesian
	20	40	1.1438	0.2784	1.1605	0.2697	1.1414 1.1356	0.1809 0.1758	1.2026 1.1932	0.2323 0.2443	E-Bayesian

**Table 6 entropy-24-01405-t006:** AEs and MSEs of the parameter $\lambda_{12}=1$ based on ELF.

n	$N_{1}$	$N_{2}$	${\hat{\lambda }}_{12MLE}$		${\hat{\lambda }}_{12BE}$		${\hat{\lambda }}_{12EBE}$		${\hat{\lambda }}_{12HBE}$		**Best Estimator**
			AE	MSE	AE	MSE	AE	MSE	AE	MSE	
40	15	15	1.1003	0.1838	1.0913	0.1027	1.0796 1.0854	0.1088 0.1079	1.1831 1.1816	0.1290 0.1248	E-Bayesian
	20	10	1.0576	0.1601	1.0912	0.1186	1.1394 1.1353	0.1290 0.1281	1.1665 1.1667	0.1238 0.1210	Bayesian
	10	20	1.1408	0.2634	1.1717	0.2732	0.9792 0.9853	0.1233 0.1176	1.1976 1.1955	0.2335 0.2300	E-Bayesian
60	23	23	1.1217	0.1944	1.1131	0.1578	1.1042 1.0997	0.1353 0.1436	1.0804 1.0911	0.1620 0.1694	H-Bayesian
	30	16	1.0917	0.11457	1.0994	0.1185	1.0213 1.0182	0.1418 0.1425	1.0812 1.0907	0.1279 0.1324	E-Bayesian
	16	30	1.1888	0.2456	1.2075	0.2338	1.1797 1.1757	0.2108 0.2086	1.1570 1.1667	0.1902 0.2003	H-Bayesian
80	30	30	1.1232	0.1987	1.1311	0.1297	1.1380 1.1347	0.1658 0.1653	1.0455 1.0448	0.1727 0.1871	H-Bayesian
	40	20	0.9247	0.1302	1.1080	0.1239	0.9740 0.9806	0.1120 0.1106	1.0234 1.0203	0.1339 0.1323	E-Bayesian
	20	40	1.1438	0.2784	1.1605	0.2697	0.9638 0.9671	0.1578 0.1654	1.1224 1.1168	0.1650 0.1901	E-Bayesian

**Table 7 entropy-24-01405-t007:** AEs and MSEs of the parameter $\lambda_{12}=1$ based on LLF.

n	$N_{1}$	$N_{2}$	${\hat{\lambda }}_{12MLE}$		${\hat{\lambda }}_{12BL}$		${\hat{\lambda }}_{12EBL}$		${\hat{\lambda }}_{12HBL}$		Best Estimator
			AE	MSE	AE	MSE	AE	MSE	AE	MSE	
40	15	15	1.1003	0.1838	1.0913	0.1027	1.09727 1.08002	0.1062 0.1123	1.1043 1.0951	0.1034 0.1067	E-Bayesian
	20	10	1.0576	0.1601	1.0912	0.1186	1.0474 1.0465	0.0862 0.0973	1.0404 1.0430	0.1041 0.0936	E-Bayesian
	10	20	1.1408	0.2634	1.1717	0.2732	0.8978 0.9012	0.1638 0.1579	1.0372 1.0348	0.1402 0.1587	H-Bayesian
60	23	23	1.1217	0.1944	1.1131	0.1578	1.1110 1.1023	0.1303 0.1294	1.0933 1.0974	0.1274 0.1017	H-Bayesian
	30	16	1.0917	0.11457	1.0994	0.1185	1.0883 1.0898	0.1401 0.1398	1.0523 1.0638	0.0992 0.1274	H-Bayesian
	16	30	1.1888	0.2456	1.2075	0.2338	1.0379 1.0381	0.2043 0.2396	1.0642 1.0778	0.2473 1.2106	E-Bayesian
80	30	30	1.1232	0.1987	1.1311	0.1297	1.1228 1.1235	0.1628 0.1579	0.9898 0.9954	0.1212 0.1174	H-Bayesian
	40	20	0.9247	0.1302	1.1080	0.1239	0.9398 0.9378	0.1079 0.1022	1.0232 1.0250	0.1103 0.1080	H-Bayesian
	20	40	1.1438	0.2784	1.1605	0.2697	0.9225 0.9279	0.1427 0.1529	1.1789 1.1829	0.2319 0.2296	E-Bayesian

**Table 8 entropy-24-01405-t008:** AEs and MSEs of the parameter $\lambda_{21}=4$ based on SELF.

n	$N_{1}$	$N_{2}$	${\hat{\lambda }}_{21MLE}$		${\hat{\lambda }}_{21BS}$		${\hat{\lambda }}_{21EBS}$		${\hat{\lambda }}_{21HBS}$		Best Estimator
			AE	MSE	AE	MSE	AE	MSE	AE	MSE	
40	15	15	4.1663	0.4725	4.1064	0.4414	4.0965 4.0858	0.3378 0.3412	4.1217 4.1218	0.4314 0.4237	E-Bayesian
	20	10	3.8763	0.3794	3.8847	0.3358	3.8802 3.8544	0.4354 0.4401	4.1941 4.1809	0.4779 0.4649	Bayesian
	10	20	3.8647	0.2439	3.9082	0.2629	3.9169 3.9129	0.3313 0.3137	4.1012 4.1035	0.4153 0.3988	E-Bayesian
60	23	23	4.1242	0.3815	4.1293	0.3983	4.0857 4.0917	0.3735 0.3621	4.0566 4.0863	0.3280 0.3312	H-Bayesian
	30	16	4.2365	0.3178	3.8594	0.2615	4.1617 4.1559	0.3239 0.3190	4.1217 4.1234	0.2613 0.2529	H-Bayesian
	16	30	3.8569	0.3102	3.8924	0.2797	3.9007 3.9021	0.2444 0.2503	4.1224 4.1306	0.3628 0.3604	E-Bayesian
80	30	30	3.7681	0.3148	3.8223	0.3125	3.8469 3.8434	0.2884 0.2935	4.1125 4.1174	0.2189 0.2108	H-Bayesian
	40	20	4.2147	0.4041	4.1542	0.3706	4.1178 4.1126	0.2516 0.2775	4.1345 4.1321	0.3605 0.3768	E-Bayesian
	20	40	3.8704	0.2463	3.8951	0.2506	3.9149 3.9184	0.3190 0.3220	4.0890 4.0927	0.3378 0.3276	H-Bayesian

**Table 9 entropy-24-01405-t009:** AEs and MSEs of the parameter $\lambda_{21}=4$ based on ELF.

n	$N_{1}$	$N_{2}$	${\hat{\lambda }}_{21MLE}$		${\hat{\lambda }}_{21BE}$		${\hat{\lambda }}_{21EBE}$		${\hat{\lambda }}_{21HBE}$		Best Estimator
			AE	MSE	AE	MSE	AE	MSE	AE	MSE	
40	15	15	4.1663	0.4725	4.1064	0.4414	4.0831 4.0788	0.2848 0.2764	4.1256 4.1168	0.3752 0.3710	E-Bayesian
	20	10	3.8763	0.3794	3.8847	0.3358	3.8530 3.8612	0.4329 0.4275	4.1303 4.1295	0.3854 0.3253	Bayesian
	10	20	3.8647	0.2439	3.9082	0.2629	3.8960 3.8994	0.1453 0.1581	4.1041 4.1019	0.2415 0.2482	E-Bayesian
60	23	23	4.1242	0.3815	4.1293	0.3983	4.0739 4.0810	0.4334 0.4130	4.0533 4.0675	0.3940 0.4003	H-Bayesian
	30	16	4.2365	0.3178	3.8594	0.2615	4.1055 4.0981	0.3432 0.3301	4.0896 4.0920	0.2761 0.2724	H-Bayesian
	16	30	3.8569	0.3102	3.8924	0.2797	3.9619 3.9565	0.2952 0.2828	4.0899 4.0882	0.3127 0.3238	E-Bayesian
80	30	30	3.7681	0.3148	3.8223	0.3125	3.8092 3.7971	0.3486 0.3340	4.0930 4.1029	0.3026 0.3248	H-Bayesian
	40	20	4.2147	0.4041	4.1542	0.3706	4.1821 4.2086	0.3515 0.3487	4.1115 4.1262	0.2442 0.2792	H-Bayesian
	20	40	3.8704	0.2463	3.8951	0.2506	3.9033 3.8924	0.2204 0.2634	4.0510 4.0479	0.2044 0.1978	H-Bayesian

**Table 10 entropy-24-01405-t010:** AEs and MSEs of the parameter $\lambda_{21}=4$ based on LLF.

n	$N_{1}$	$N_{2}$	${\hat{\lambda }}_{21MLE}$		${\hat{\lambda }}_{21BL}$		${\hat{\lambda }}_{21EBL}$		${\hat{\lambda }}_{21HBL}$		Best Estimator
			AE	MSE	AE	MSE	AE	MSE	AE	MSE	
40	15	15	4.1663	0.4725	4.1064	0.4414	4.0948 4.0960	0.3552 0.3541	4.1424 4.1470	0.3932 0.4009	E-Bayesian
	20	10	3.8763	0.3794	3.8847	0.3358	3.8703 3.8867	0.4353 0.3922	4.1580 4.1515	0.3646 0.3579	Bayesian
	10	20	3.8647	0.2439	3.9082	0.2629	3.9151 3.9188	0.2539 0.2558	4.1147 4.1088	0.3356 0.3218	E-Bayesian
60	23	23	4.1242	0.3815	4.1293	0.3983	4.0996 4.1078	0.2940 0.2749	4.1030 4.0975	0.3003 0.3187	E-Bayesian
	30	16	4.2365	0.3178	3.8594	0.2615	4.1382 4.1449	0.2248 0.2380	4.1208 4.1252	0.2197 0.2119	H-Bayesian
	16	30	3.8569	0.3102	3.8924	0.2797	3.9138 3.9165	0.2248 0.2527	4.0941 4.0897	0.3228 0.3340	E-Bayesian
80	30	30	3.7681	0.3148	3.8223	0.3125	3.8463 3.8452	0.3203 0.3312	4.1051 4.1046	0.2732 0.2825	H-Bayesian
	40	20	4.2147	0.4041	4.1542	0.3706	4.1073 4.1154	0.2754 0.2831	4.1343 4.1430	0.3541 0.3306	E-Bayesian
	20	40	3.8704	0.2463	3.8951	0.2506	3.9249 3.9211	0.2810 0.2717	4.0749 4.0672	0.1620 0.1521	H-Bayesian

**Table 11 entropy-24-01405-t011:** AEs and MSEs of the parameter $\lambda_{22}=2$ based on SELF.

n	$N_{1}$	$N_{2}$	${\hat{\lambda }}_{22MLE}$		${\hat{\lambda }}_{22BS}$		${\hat{\lambda }}_{22EBS}$		${\hat{\lambda }}_{22HBS}$		Best Estimator
			AE	MSE	AE	MSE	AE	MSE	AE	MSE	
40	15	15	1.8896	0.2261	2.1012	0.2662	1.9012 1.9065	0.1923 0.1713	2.1217 2.1315	0.3897 0.3405	E-Bayesian
	20	10	1.8426	0.3504	1.8458	0.3216	1.8361 1.8327	0.3661 0.3621	2.1612 2.1586	0.4151 0.4164	Bayesian
	10	20	2.1361	0.2964	1.9044	0.2371	2.0502 2.0318	0.1874 0.1916	2.0923 2.0982	0.3362 0.3417	E-Bayesian
60	23	23	2.1481	0.2030	2.1477	0.2242	1.8998 1.9050	0.2087 0.1961	2.1033 2.1027	0.2473 0.2523	E-Bayesian
	30	16	2.1651	0.2893	2.1778	0.3002	2.1414 2.1224	0.2579 0.2637	2.1374 2.1366	0.2225 0.2352	H-Bayesian
	16	30	2.1330	0.2160	2.1237	0.1881	1.9286 1.9244	0.2212 0.2098	2.0864 2.0758	0.2974 0.2736	E-Bayesian
80	30	30	2.1331	0.1842	2.1388	0.1667	2.1481 2.1373	0.1644 0.1693	2.1038 2.1082	0.1346 0.1338	H-Bayesian
	40	20	2.1878	0.2935	2.1801	0.3034	2.1626 2.1708	0.3323 0.3105	2.1826 2.1756	0.3526 0.3781	E-Bayesian
	20	40	2.1176	0.2165	2.1208	0.1508	2.1201 2.1188	0.1237 0.1200	2.0438 2.0386	0.1078 0.1056	H-Bayesian

**Table 12 entropy-24-01405-t012:** AEs and MSEs of the parameter $\lambda_{22}=2$ based on ELF.

n	$N_{1}$	$N_{2}$	${\hat{\lambda }}_{22MLE}$		${\hat{\lambda }}_{22BE}$		${\hat{\lambda }}_{22EBE}$		${\hat{\lambda }}_{22HBE}$		Best Estimator
			AE	MSE	AE	MSE	AE	MSE	AE	MSE	
40	15	15	1.8896	0.2261	2.1012	0.2662	1.9159 1.9019	0.2176 0.2136	2.1008 2.1011	0.2928 0.2613	E-Bayesian
	20	10	1.8426	0.3504	1.8458	0.3216	1.9088 1.9027	0.2637 0.2597	2.1181 2.1202	0.3377 0.3217	E-Bayesian
	10	20	2.1361	0.2964	1.9044	0.2371	2.0558 2.0582	0.2088 2.2134	2.0945 2.0979	0.2798 0.2808	E-Bayesian
60	23	23	2.1481	0.2030	2.1477	0.1942	1.9179 1.9043	0.1815 0.1696	2.0919 2.0902	0.1577 0.1519	H-Bayesian
	30	16	2.1651	0.2893	2.1778	0.3002	1.9852 1.9676	0.2918 0.3069	2.1355 2.1595	0.3353 0.3459	E-Bayesian
	16	30	2.1330	0.2160	2.1237	0.1881	1.8873 1.8942	0.1987 0.1879	2.0558 2.0562	0.2434 0.2106	H-Bayesian
80	30	30	2.1331	0.1842	2.1388	0.1667	2.1234 2.1207	0.1640 0.1629	2.1025 2.1085	0.1493 0.1342	H-Bayesian
	40	20	2.1878	0.2935	2.1801	0.3034	2.1570 2.1468	0.2119 0.2117	2.1654 2.1635	0.2360 0.2387	E-Bayesian
	20	40	2.1176	0.2165	2.1208	0.1508	2.0884 2.0981	0.2137 0.2104	1.9608 1.9799	0.1663 0.1648	H-Bayesian

**Table 13 entropy-24-01405-t013:** AEs and MSEs of the parameter $\lambda_{22}=2$ based on LLF.

n	$N_{1}$	$N_{2}$	${\hat{\lambda }}_{22MLE}$		${\hat{\lambda }}_{22BL}$		${\hat{\lambda }}_{22EBL}$		${\hat{\lambda }}_{22HBL}$		Best Estimator
			AE	MSE	AE	MSE	AE	MSE	AE	MSE	
40	15	15	1.8896	0.2261	2.1012	0.2662	1.9385 1.9249	0.1966 0.2057	1.9063 1.9148	0.3635 0.3901	E-Bayesian
	20	10	1.8426	0.3504	1.8458	0.3216	1.9036 1.9161	0.2558 0.2519	1.8968 1.8983	0.3863 0.3866	E-Bayesian
	10	20	2.1361	0.2964	1.9044	0.2371	2.1150 2.1271	0.2923 0.3013	1.9023 1.9096	0.3463 0.3384	Bayesian
60	23	23	2.1481	0.2030	2.1477	0.1942	1.8912 1.9080	0.2585 0.2474	1.9187 1.9203	0.1410 0.1373	H-Bayesian
	30	16	2.1651	0.2893	2.1778	0.3002	1.8708 1.8839	0.3696 0.3570	1.9764 1.9814	0.2338 0.2282	H-Bayesian
	16	30	2.1330	0.2160	2.1237	0.1881	1.9617 1.9489	0.1778 0.1675	1.9212 1.9334	0.2302 0.2123	E-Bayesian
80	30	30	2.1331	0.1842	2.1388	0.1667	2.0855 2.0885	0.1918 0.1907	2.1242 2.1425	0.1860 0.1845	E-Bayesian
	40	20	2.1878	0.2935	2.1801	0.3034	2.1177 2.1084	0.2436 0.2254	1.8713 1.8687	0.3313 0.3436	E-Bayesian
	20	40	2.1176	0.2165	2.1208	0.1508	2.0792 2.0732	0.2301 0.2437	1.9387 19403	0.1172 0.1137	H-Bayesian

**Table 14 entropy-24-01405-t014:** AL and CP of 95% asymptotic CIs of the parameter $\lambda_{11}$ based on 1000 replications.

n	$N_{1}$	$N_{2}$	MLE	Boot-p	Boot-t	Bay	E-Bay1	E-Bay2	H-Bay1	H-Bay2
40	15	15	2.2650 96.1	1.6267 96.2	1.6045 96.3	1.5098 96.3	1.5104 96.5	1.5087 96.5	1.6801 96.7	1.6646 96.8
	20	10	1.7281 96.4	1.5522 96.5	1.5475 96.5	1.4935 96.6	1.5012 96.8	1.5109 96.7	1.6382 96.6	1.6262 96.5
	10	20	2.3154 96.1	1.6929 96.3	1.6869 96.2	1.55720 96.2	1.5467 96.5	1.5523 96.4	1.5128 96.6	1.5057 96.7
60	23	23	1.7720 96.1	1.5697 96.3	1.6001 96.2	1.4564 96.5	1.5028 96.6	1.4875 96.6	1.5016 96.5	1.5405 96.4
	30	16	1.6327 96.4	1.5979 96.5	1.6007 96.4	1.4209 96.8	1.3883 97.0	1.4078 96.9	1.2468 97.0	1.2480 97.1
	16	30	2.0635 96.0	1.8008 96.2	1.7902 96.1	1.5702 96.3	1.6056 96.4	1.6123 96.5	1.6058 96.3	1.5735 96.3
80	30	30	1.5828 96.3	1.6270 96.6	1.6353 96.7	1.5828 96.8	1.5730 96.7	1.5854 96.8	1.3449 97.2	1.3156 97.3
	40	20	1.3788 96.8	1.5905 97.0	1.6032 96.9	1.4168 97.1	1.4140 97.0	1.3977 96.9	1.2802 97.4	1.2919 97.5
	20	40	1.8671 96.2	1.6587 96.4	1.6462 96.6	1.6249 96.6	1.5897 96.5	1.6117 96.6	1.4772 97.0	1.5190 97.1

**Table 15 entropy-24-01405-t015:** AL and CP of 95% asymptotic CIs of the parameter $\lambda_{12}$ based on 1000 replications.

n	$N_{1}$	$N_{2}$	MLE	Boot-p	Boot-t	Bay	E-Bay1	E-Bay2	H-Bay1	H-Bay2
40	15	15	1.6723 96.4	1.4043 96.6	1.4103 96.5	1.3593 96.8	1.4187 96.8	1.4298 96.7	1.3932 96.6	1.4437 96.7
	20	10	1.3693 96.6	1.3354 96.7	1.3168 96.7	1.1784 97.2	1.2657 97.0	1.2701 96.9	1.2939 96.8	1.3024 96.9
	10	20	1.7863 96.3	1.6423 96.4	1.5858 96.3	1.4460 96.6	1.3334 96.9	1.3385 96.8	1.4606 96.6	1.4642 96.6
60	23	23	1.2831 96.4	1.1406 96.5	1.1318 96.6	1.2003 96.7	1.1975 97.1	1.1893 97.0	1.2199 97.1	1.2108 96.9
	30	16	1.1871 96.5	1.1650 96.7	1.1745 96.8	1.0954 96.9	1.0876 97.3	1.0933 97.4	1.1456 97.3	1.1554 97.2
	16	30	1.5851 96.4	1.2345 96.6	1.2305 96.7	1.2202 96.6	1.2001 96.8	1.1987 96.9	1.3907 96.9	1.3896 96.8
80	30	30	1.0901 96.7	1.1267 96.8	1.1197 96.9	1.0956 97.2	1.1063 97.1	1.1085 97.0	1.0688 97.2	1.0693 97.3
	40	20	0.9497 96.8	1.0902 97.1	1.0790 97.2	1.0812 97.4	1.0743 97.3	1.0756 97.3	0.9454 97.4	0.9588 97.4
	20	40	1.3015 96.5	1.1716 96.5	1.1569 96.7	1.1277 97.1	1.1376 97.0	1.1297 96.9	1.3364 97.0	1.3992 97.1

**Table 16 entropy-24-01405-t016:** AL and CP of 95% asymptotic CIs of the parameter $\lambda_{21}$ based on 1000 replications.

n	$N_{1}$	$N_{2}$	MLE	Boot-p	Boot-t	Bay	E-Bay1	E-Bay2	H-Bay1	H-Bay2
40	15	15	4.6865 95.6	3.4313 95.8	3.4951 95.9	2.8982 96.1	2.7869 96.0	2.8065 95.9	3.0162 95.7	3.0273 95.7
	20	10	3.7323 95.8	3.0256 96.0	3.0088 96.0	2.3586 96.2	2.4067 96.1	2.4198 96.1	3.4455 95.8	3.1465 95.7
	10	20	3.2754 95.9	2.9521 96.2	2.9771 96.1	2.1729 96.3	2.2471 96.2	2.3089 96.2	2.8502 95.9	2.8535 96.0
60	23	23	3.6721 96.2	3.3443 96.3	3.3672 96.5	2.6876 96.4	2.7013 96.7	2.4987 96.8	2.8644 96.7	2.7576 96.8
	30	16	4.4871 96.0	3.4678 96.2	3.5389 96.3	2.6006 96.4	2.5978 96.6	2.5409 96.6	2.9116 96.5	2.9149 96.5
	16	30	3.0036 96.4	3.0457 96.5	3.0056 96.7	2.4792 96.6	2.385 96.9	2.455 97.0	2.4469 97.0	2.4607 97.1
80	30	30	2.8167 96.5	2.9564 96.5	2.8902 96.7	2.8167 96.8	2.8637 96.9	2.8232 97.0	2.3535 97.2	2.2528 97.3
	40	20	3.0092 96.4	3.0532 96.4	3.1671 96.5	2.9869 96.7	3.0011 96.7	2.9945 96.8	2.8054 96.9	2.7909 97.1
	20	40	2.5920 96.6	2.8737 96.9	2.7877 97.1	2.6943 97.2	2.7089 97.3	2.6880 97.3	2.2513 97.4	2.1863 97.5

**Table 17 entropy-24-01405-t017:** AL and CP of 95% asymptotic CIs of the parameter $\lambda_{22}$ based on 1000 replications.

n	$N_{1}$	$N_{2}$	MLE	Boot-p	Boot-t	Bay	E-Bay1	E-Bay2	H-Bay1	H-Bay2
40	15	15	2.7919 95.2	2.0592 96.4	2.0366 96.3	1.9583 96.5	2.2453 96.3	2.2567 96.3	2.2574 96.2	2.2464 96.1
	20	10	2.8595 95.1	2.3604 95.4	2.3537 95.8	2.3724 96.0	2.3434 96.2	2.3512 96.2	2.5667 96.0	26456 95.9
	10	20	2.4929 95.4	2.2789 95.7	2.2786 95.8	2.4052 95.9	2.1564 96.5	2.1677 96.4	2.2062 96.2	2.1921 96.3
60	23	23	2.526 96.4	2.2596 96.5	2.2972 96.7	1.9104 96.9	1.9234 96.6	1.9097 96.6	2.0271 96.7	2.0414 96.6
	30	16	2.6553 96.2	2.5979 96.4	2.5328 96.5	2.0071 96.7	2.0198 96.8	2.0210 96.7	1.9900 96.8	2.0108 96.7
	16	30	2.3803 96.6	2.2180 96.7	2.2275 96.8	1.8775 96.9	1.9034 96.9	1.8967 96.8	1.8607 97.1	1.8715 97.0
80	30	30	2.2274 96.7	2.2970 96.8	2.3097 96.7	2.2273 97.1	2.2456 96.9	2.2428 96.8	2.0538 97.2	2.0419 97.0
	40	20	2.6615 96.2	2.3904 96.4	2.3464 96.4	2.2612 96.7	2.3014 96.4	2.2951 96.5	2.1447 96.7	2.1797 96.8
	20	40	1.9461 96.8	2.2842 96.9	2.2096 97.0	2.1535 97.4	2.1837 97.3	2.1756 97.3	1.8597 97.4	1.7808 97.5

**Table 18 entropy-24-01405-t018:** AL and CP of 95% HPD CIs of the parameter $\lambda_{11}$ based on 1000 replications.

n	$N_{1}$	$N_{2}$	Bay	E-Bay1	E-Bay2	H-Bay1	H-Bay2
40	15	15	1.7864 96.7	1.6341 96.8	1.6402 96.8	1.7754 96.7	1.8354 96.7
	20	10	1.6324 96.8	1.5776 96.9	1.5809 97.0	1.6393 96.9	1.6238 96.9
	10	20	1.8721 96.8	1.9467 96.7	1.9387 96.7	2.0227 96.6	2.0178 96.6
60	23	23	1.5403 96.7	1.4650 96.8	1.4701 96.7	1.4806 96.6	1.4776 96.6
	30	16	1.3917 96.9	1.2199 97.0	1.2740 97.1	1.3492 96.8	1.3406 96.7
	16	30	1.6362 96.5	1.5267 96.7	1.5249 96.8	1.4258 97.1	1.4375 97.1
80	30	30	1.3275 96.7	1.3974 96,6	1.2896 96.7	1.2661 97.2	1.2470 97.4
	40	20	1.1935 97.1	1.1890 97.2	1.2011 97.0	1.1279 97.5	1.1489 97.6
	20	40	1.5047 96.6	1.5104 96.6	1.5098 96.5	1.3785 97.2	1.3088 97.3

**Table 19 entropy-24-01405-t019:** AL and CP of 95% HPD CIs of the parameter $\lambda_{12}$ based on 1000 replications.

n	$N_{1}$	$N_{2}$	Bay	E-Bay1	E-Bay2	H-Bay1	H-Bay2
40	15	15	1.2443 96.2	1.2067 96.5	1.1999 96.8	1.1934 96.9	1.1662 97.0
	20	10	1.1185 96.6	1.1367 96.8	1.1289 97.0	1.0630 97.0	1.0445 97.1
	10	20	1.3060 96.0	1.2567 96.4	1.2481 96.7	1.2342 96.8	1.1994 96.9
60	23	23	1.1831 96.7	1.0165 96.9	1.0189 96.8	1.1312 97.0	1.1338 97.1
	30	16	1.1788 96.9	0.8824 97.1	0.8743 97.2	1.0824 97.1	1.0943 97.2
	16	30	1.1977 96.7	1.1876 96.9	1.1901 96.8	1.1562 97.0	1.1679 97.1
80	30	30	1.0754 96.9	1.0854 97.0	1.0812 97.1	1.1258 97.1	1.1514 97.3
	40	20	1.0660 97.2	1.0667 97.1	1.0589 97.2	1.0289 97.2	1.0174 97.5
	20	40	1.1099 96.8	1.1123 96.9	1.1143 97.0	1.1638 97.1	1.1769 97.2

**Table 20 entropy-24-01405-t020:** AL and CP of 95% HPD CIs of the parameter $\lambda_{21}$ based on 1000 replications.

n	$N_{1}$	$N_{2}$	Bay	E-Bay1	E-Bay2	H-Bay1	H-Bay2
40	15	15	2.2610 96.7	2.2089 96.8	2.1998 96.9	2.4482 96.7	2.4560 96.7
	20	10	2.3346 96.6	2.4067 96.6	2.4145 96.5	2.5248 96.4	2.5436 96.3
	10	20	2.2429 96.8	2.1698 97.2	2.1704 97.3	2.2213 97.1	2.2357 97.0
60	23	23	2.3493 96.5	2.2719 96.9	2.2987 96.8	2.2024 97.0	2.1932 97.1
	30	16	2.3732 96.3	2.3545 96.8	2.3499 96.7	2.2676 96.9	2.2569 96.9
	16	30	2.2214 96.8	2.1978 97.0	2.2054 97.0	2.0785 97.3	2.0804 97.4
80	30	30	2.3141 96.7	2.2270 97.0	2.2320 97.1	2.1084 97.4	1.9397 97.6
	40	20	2.4359 96.4	2.3976 96.9	2.4012 96.9	2.3392 97.3	2.3285 97.4
	20	40	2.1551 96.9	2.1485 97.2	2.1432 97.3	2.0285 97.5	1.9038 97.7

**Table 21 entropy-24-01405-t021:** AL and CP of 95% HPD CIs of the parameter $\lambda_{22}$ based on 1000 replications.

n	$N_{1}$	$N_{2}$	Bay	E-Bay1	E-Bay2	H-Bay1	H-Bay2
40	15	15	1.8864 96.5	1.6887 96.8	1.7014 96.7	2.1704 95.9	2.1615 96.0
	20	10	2.4531 95.3	2.2454 95.7	2.3089 96.0	2.3015 95.7	2.2925 95.6
	10	20	2.0525 96.2	1.9001 96.6	1.8976 96.6	2.0185 96.3	1.9726 96.4
60	23	23	2.5403 96.1	2.3867 96.1	2.4009 96.0	1.8808 96.6	1.8783 96.8
	30	16	2.6916 95.9	2.4563 96.0	2.5012 95.9	1.9384 96.5	1.9467 96.7
	16	30	2.3362 96.2	2.1554 96.4	2.1398 96.5	1.6731 96.9	1.6643 97.0
80	30	30	1.6275 97.0	1.6267 96.9	1.6358 96.8	1.5221 97.1	1.5097 97.2
	40	20	2.2935 96.4	2.1152 96.6	2.0147 96.6	1.8577 96.8	1.8449 96.9
	20	40	1.5047 97.2	1.5107 97.3	1.5132 97.4	1.4590 97.6	1.4424 97.7

### 7.2. Results Analysis

From Table 2, Table 3, Table 4, Table 5, Table 6, Table 7, Table 8, Table 9, Table 10, Table 11, Table 12, Table 13, Table 14, Table 15, Table 16, Table 17, Table 18, Table 19, Table 20 and Table 21, some conclusions are summarized as follows:
(1)From Table 2, Table 3, Table 4, Table 5, Table 6, Table 7, Table 8, Table 9, Table 10, Table 11, Table 12 and Table 13, we observe that the AEs of $\lambda_{ij}(i,j=1,2)$ are close to the true values and the MSEs of $\lambda_{ij}(i,j=1,2)$ decrease as n increasing for all estimates. This indicates that the number of failure values of test units affect the estimation accuracy of parameters.(2)From Table 2, Table 3, Table 4, Table 5, Table 6, Table 7, Table 8, Table 9, Table 10, Table 11, Table 12 and Table 13, the Bayesian performances are better than that of MLE, and the E-Bayesian or H-Bayesian performances are better than that of the Bayesian for fixed n, $N_{1}$, $N_{2}$, and censoring scheme. The results show that the Bayesian method improves the estimation accuracy of model parameters due to combining the prior information.(3)From Table 2, Table 3, Table 4, Table 5, Table 6, Table 7, Table 8, Table 9, Table 10, Table 11, Table 12 and Table 13, we can infer that the H-BEs are the best in all cases of the larger sample sizes, and the E-BEs are the best in all cases of the smaller sample sizes based on different loss functions.(4)From Table 2, Table 3, Table 4, Table 5, Table 6, Table 7, Table 8, Table 9, Table 10, Table 11, Table 12 and Table 13, we observe that the proportion of failure values under the stress $S_{1}$ is greater than the one under the stress $S_{2}$, the estimated values of the parameters $\lambda_{1j}(j=1,2)$ are close to the true values, and vice versa. This shows that the pre-fixed time τ in the test also affects the estimation accuracy of model parameters.(5)From Table 14, Table 15, Table 16, Table 17, Table 18, Table 19, Table 20 and Table 21, the ALs of all asymptotic CRIs and HPDCRIs become smaller, and the CPs are very close to the corresponding nominal level as n increases.(6)From Table 14, Table 15, Table 16, Table 17, Table 18, Table 19, Table 20 and Table 21, we observe that the H-Bayesian CIs are always narrower than the other CIs and the HPDCRIs are always narrower than the other CIs under the same loss function.

## 8. An Illustrative Example

In this section, we simulate a PT-IIC sample from a simple step-stress accelerated competing failure model. The dataset is generated with the following choices of the parameters: $\lambda_{11}=1.0$, $\lambda_{12}=1.5$, $\lambda_{21}=2.0$ and $\lambda_{22}=3.0$, and n=30, $N_{1}=10$, $R_{1}=4$, $N_{2}=12$, and $R_{2}=4$. The data are given in Table 22. From this dataset, we have $n_{11}=4$, $n_{12}=6$, $n_{21}=5$ and $n_{22}=7$, and the Average Estimates (AEs) of MLEs, BEs, EBs, and HBs of the parameters are derived based on the SELF. The results are presented in Table 23. From Table 23, it is clearly observed that the E-Bayesian or H-Bayesian performances are better than the MLEs.

We constructed the ALs of the 95% asymptotic CIs and the results are presented in Table 24. We also consider the HPDCRIs, and the results are presented in Table 25. From Table 24 and Table 25, we observe that the H-Bayesian CIs are always narrower than the other CIs. The HPDCRIs are always narrower than the other CIs.

**Table 22 entropy-24-01405-t022:** The data for an illustrative example.

**First Stress Level**	(0.00638,2), (0.01442,1), (0.01738,1), (0.02380,2), (0.04067,2), (0.05375,2), (0.06667,1), (0.08122,1), (0.11568,2), (0.15354,2)
**Second Stress Level**	(0.17226,1), (0.18334,2), (0.20501,2), (0.21434,1), (0.21518,2), (0.22165,1), (0.23910,1), (0.24391,1), (0.26104,2), (0.32582,2), (0.34505,2), (0.65557,2)

**Table 23 entropy-24-01405-t023:** AEs of the parameters for an illustrative example.

Parameter	MLE	BE	EBE1	EBE2	HBE1	HBE2
$\lambda_{11}$	1.09	0.968	1.02	0.966	1.016	1.021
$\lambda_{12}$	1.62	1.516	1.47	1.485	1.511	1.520
$\lambda_{22}$	3.21	2.83	2.92	2.894	3.052	3.092

**Table 24 entropy-24-01405-t024:** The ALs of 95% asymptotic CIs for an illustrative example.

Parameter	MLE	BE	EBE1	EBE2	HBE1	HBE2
$\lambda_{11}$	2.127	1.709	1.712	1.699	1.377	1.409
$\lambda_{12}$	2.605	2.194	2.201	2.217	1.514	1.521
$\lambda_{21}$	3.529	3.025	3.125	3.221	2.546	2.691
$\lambda_{22}$	3.268	2.997	3.009	3.014	1.795	1.802

**Table 25 entropy-24-01405-t025:** The ALs of 95% HPD CIs for an illustrative example.

Parameter	BE	EBE1	EBE2	HBE1	HBE2
$\lambda_{11}$	1.621	1.635	1.659	1.346	1.361
$\lambda_{12}$	2.158	2.098	2.117	1.481	1.457
$\lambda_{21}$	2.807	2.765	2.769	2.184	2.201
$\lambda_{22}$	2.679	2.544	2.608	1.685	1.595

## 9. Conclusions

This paper has proposed a simple S-SALT based on the CEM and under different PT-IIC schemes. It has been assumed that the lifetimes have an exponential distribution with different scale parameters. We have derived the maximum likelihood estimations, Bayesian estimations, Expected Bayesian estimations, and Hierarchical Bayesian estimations of the scale parameters based on the different LF. Based on Monte Carlo Simulations, we also obtained the ALs and the CPs of the 95% ACIs, BPCIs, BTCIs, and HPDCRIs for all the unknown parameters. Results show that the MSEs of the scale parameters decrease as the experimental units N increasing for all estimators, the H-BEs are the best in all cases of the larger samples sizes, and the E-BEs are the best in all cases of the smaller samples sizes. From the perspective of the CIs and CRIs, it has been observed that the H-Bayesian CIs are always narrower than the other Cis, and the HPDCRIs are always narrower than the other CIs under the same loss function. Note that in this paper we assume that the two competing failures are independent. In future work, the simple step-stress accelerated dependent competing failure model will be considered, such as Copula [37,38], which is one of the popular models for releasing the restriction.

## Data Availability

The data are available from the corresponding author upon request.

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
