# Peer review of "E-Bayesian and H-Bayesian Inferences for a Simple Step-Stress Model with Competing Failure Model under Progressively Type-II Censoring"

_entropy, 2022, doi:10.3390/e24101405_

Round 1

Reviewer 1 Report

This is a nice article which "turns the Bayesian crank" for an interesting experimental design related to failure rate modeling.  I have no serious objections to the material presented and agree that is a nice result which should be of interest to the readership of this journal.  A few commets:

(1) The underlying failure model the authors assume is relatively simple, essentially an exponential model which an linear inverse relationship on the exponential parameter and the input stress.  The authors might consider spending some time discussing the shortcomings of this framework and investigating how their Bayesian formulation performs when the true model deviates from this

(2)  The results in this article appear to be solely from simulation studies which are "on the model", i.e. generated from the model they have specified.  In this case, it is unsurprising that the Bayesian approach outperforms.  It would again be interesting to see how this transfers over to model misspecified situations.

(3)  I found the exposition in the beginning of Secton 2.1 confusing.  It would be helpful if the authors rewrote things a bit.  For example, delta, is not explained until the end of the paragraph, which makes the entire development hard to follow.  Similarly, L is, as far as I can tell, never defined.

(4)  It would have been interesting to see this model applied to an actual dataset

In general, the derivations seem correct and the results are encouraging.  It would be nice to clean up the exposition in section 2.1 a bit and perhaps provide a bit more context

Reviewer 2 Report

This paper develops Bayesian and MLE-based inference on the simple step-stress model. Extensive simulations were conducted to verify the performance of the propose method.

Main concerns:

1. I do not suggest using the abbreviation CFM that is not common in the literature and difficult to read.

2. Some equations are unreadable due to bugs. After "the remaining" in P.2, So many bugs in the formulas of P.5,

 shown as "L" and "M". Formulas in the titles of Tables mostly include bugs.

 Consequently, the paper becomes unreadable at this stage.

3. The paper needs to undergo a strict proof reading since there are lots of errors and unnatural wordings.

 English must be sophisticated and corrected. I write some of suggestions below, but there are lots of others.

P.1, Some references in the field --> Some authentic books in the field

P.1, According inference the constant-stress --> The constant-stress

P.1, to analyze the modeling data --> to analyze the data

P.2, The unit failure only --> The unit fails only

P.23, "are closed to" --> "are close to"

4. As the authors correctly acknowledge in the conclusion, the independence assumption of failures T1 and T2 are 

unrealistic. Therefore, it is best to suggest useful models for dependent competing risks, such as copula models [1,2,3],

 shortly at the end of the section.

[1] Escarela, G., & Carriere, J. F. (2003). Fitting competing risks with an assumed copula.

 Statistical Methods in Medical Research, 12(4), 333-349.

[2] Michimae H, Emura T (2022) Likelihood inference for copula models based on left-truncated and competing

 risks data from field studies, Mathematics 10(13):2163

[3] Wang, Y. C., Emura, T., Fan, T. H., Lo, S. M., & Wilke, R. A. (2020). Likelihood‐based inference for a frailty‐copula model

 based on competing risks failure time data. Quality and Reliability Engineering International, 36(5), 1622-1638.

5. I am disappointed to see the artificial data for the real data analysis.

 This means that there is no any dataset that the proposed method can apply, and the method

 is merely imaginary.

6. Some names of the journals in the references are not correct. For instannce, use this : "Commun. Stat. - Theory Methods".

Reviewer 3 Report

Please see the attached comments.

Round 2

Reviewer 2 Report

The authors suffiiently addressed the concerns in my previosu reivew report, though my concern for the real data analysis was not addressed.

 I am still suggesting the paper to be acceptable in its present form.

Author Response

Thank you very much for the reviewer's approval of the modification, and the real data analysis will continue in future research.

Reviewer 3 Report

In the attached document, I have repeated the points which have not been fully addressed and added my responses to your responses.

Round 3

Reviewer 3 Report

Comments are provided in the attached file.
